# Generating ring-shaped engineered heart tissues from ventricular and atrial human pluripotent stem cell-derived cardiomyocytes

Idit Goldfracht [1], Stephanie Protze[2,3], Assad Shiti[1], Noga Setter [1], Amit Gruber[1], Naim Shaheen[1], Yulia Nartiss[2], Gordon Keller[2,4,5] & Lior Gepstein[1,6]*

The functions of the heart are achieved through coordination of different cardiac cell sub-types (e.g., ventricular, atrial, conduction-tissue cardiomyocytes). Human pluripotent stem cell-derived cardiomyocytes (hPSC-CMs) offer unique opportunities for cardiac research. Traditional studies using these cells focused on single-cells and utilized mixed cell populations. Our goal was to develop clinically-relevant engineered heart tissues (EHTs) comprised of chamber-specific hPSC-CMs. Here we show that such EHTs can be generated by directing hPSCs to differentiate into ventricular or atrial cardiomyocytes, and then embedding these cardiomyocytes in a collagen-hydrogel to create chamber-specific, ring-shaped, EHTs. The chamber-specific EHTs display distinct atrial versus ventricular phenotypes as revealed by immunostaining, gene-expression, optical assessment of action-potentials and conduction velocity, pharmacology, and mechanical force measurements. We also establish an atrial EHT-based arrhythmia model and confirm its usefulness by applying relevant pharmacological interventions. Thus, our chamber-specific EHT models can be used for cardiac disease modeling, pathophysiological studies and drug testing.

[1] Sohnis Research Laboratory for Cardiac Electrophysiology and Regenerative Medicine, the Rappaport Faculty of Medicine and Research Institute, Technion–Israel Institute of Technology, Haifa, Israel, POB 9649, Haifa 3109601, Israel. [2] McEwen Stem Cell Institute, University Health Network, Toronto M5G 1L7 Ontario, Canada. [3] Department of Molecular Genetics, University of Toronto, Toronto, ON M5S 1A8, Canada. [4] Department of Medical Biophysics, University of Toronto, Toronto, ON M5G 1L7, Canada. [5] Princess Margaret Cancer Center, University Health Network, Toronto, ON M5G 1L7, Canada. [6] Cardiology Department, Rambam Health Care Campus, Haliya Hashniya St 8, Haifa 3109601, Israel. *email: mdlior@technion.ac.il

The main functions of the heart are achieved by the coordinated activity of different cell types, such as ventricular, atrial, or conduction-system cardiomyocytes[1] as well as non-myocytes. Human pluripotent stem cell-derived cardiomyocytes (hPSC-CMs) offer several opportunities for cardiovascular regenerative medicine[2–7], for studying human heart development[8–10], for modeling acquired[11,12] and inherited[13–16] cardiac disorders, and for studying the effects of drugs and other therapeutic interventions[16,17]. Most of these studies, however, focused on the single-cell level and utilized hPSC-CMs populations consisting of a mixture of different cardiomyocyte subtypes, such as ventricular-, atrial-, and nodal-like cells[13,15,18,19]. This cellular heterogeneity may significantly hamper the use of hPSC-CMs for many of the aforementioned applications. To address this challenge, significant efforts have been made recently to establish development biology-guided hPSC differentiation protocols and/or selection strategies in an attempt to generate purified populations of chamber-specific cardiomyocytes[8,10,20–23].

Cardiac tissue engineering is an emerging biomedicine discipline attempting to combine scaffolding polymers with cells to create cardiac muscle like tissue-constructs[24,25]. Previous studies with engineered heart tissues (EHT) composed of hPSC-CMs demonstrated that they resemble the native heart[26–28] and can be used for studying basic features of myocardial tissue biology[27–29], developing in-vitro models of acquired[29,30] and inherited[31–33] cardiac disorders, drug screening[32–36], and for myocardial tissue-replacement therapies[37–39]. Although these tissue models proved useful for the aforementioned applications, they have also been hampered by the heterogeneity of their cellular composition. Consequentially, one of the important challenges in the field of hPSC-based cardiac tissue engineering is deriving chamber-specific myocardial tissues comprised purified populations of specific hPSC-CM subtypes.

Here, we aim to address this challenge and to develop methods for deriving hPSC-based ventricular and atrial engineered tissues, to characterize their phenotypic properties, and to evaluate the impact of such tissue models for downstream applications. Using a development biology-guided differentiation approach[8], we generate purified populations of either ventricular- or atrial-like cells. The hPSC-CM subtypes are then combined with collagen to create chamber-specific EHTs. By combing immunostaining, laser-confocal functional imaging, optical mapping, force measurements, and targeted pharmacology, we can identify clear and distinguishable differences in the phenotypic properties of the ventricular and atrial engineered tissues. We then demonstrate the potential of such chamber-specific cardiac-tissue models for various physiological studies, disease modeling, and drug testing applications.

## Results

### Differentiation into ventricular and atrial cardiomyocytes.
Human embryonic stem cells (hESC) were differentiated into ventricular- or atrial-like cells using protocols based on the sequential manipulation of the BMP, activin-nodal, Wnt, and retinoic-acid (RA) signaling pathways (Fig. 1a)[8,9]. The main differences in the ventricular versus atrial differentiation protocols included variation in the concentrations of activin A and BMP4 at day 1 of differentiation (d1) for induction of atrial or ventricular mesoderm and the addition of retinoic-acid (RA, 0.25–0.5 μM) on d3 to induce differentiation of atrial cardiomyocytes. These differentiation schemes yielded spontaneous beating cardiomyocytes, whose chamber-specific phenotype could already be appreciated by their different beating patterns; slower and more vigorous contraction of the ventricular-like cells

compared with the faster contraction of atrial cells (Supplementary Movie 1).

The HES3-NKX2–5[egfp/w] reporter hESC line was used to monitor cardiomyocyte differentiation. Flow-cytometry analysis for eGFP (identifying NKX2–5-expressing cells) and the cardiac-specific marker cardiac troponin T (cTnT) on d20 confirmed the efficiency of both ventricular and atrial differentiation protocols, resulting in $88 \pm 1\%$ ($n = 30$) and $82 \pm 2\%$ ($n = 34$) eGFP$^+$ and cTnT$^+$ cardiomyocytes, respectively (Fig. 1b, c, left panels, values are expressed as mean ± SEM). We next evaluated the percentage of cTnT$^+$/NKX2.5$^+$ cells that also express MLC-2v, which is exclusively expressed in ventricular cardiomyocytes throughout development and is absent in atrial cells[1,8]. Flow-cytometry analysis (Fig. 1b, c, right panels) revealed that the vast majority of cTnT$^+$/NKX2.5$^+$ cells derived using the ventricular differentiation protocol were MLC-2v$^+$, whereas fewer than 5% of the differentiating cardiomyocytes obtained using the atrial-specific protocol expressed this marker (Fig. 1b, c, right panels). To further validate the ventricular-specific differentiation protocol, we plated the differentiating hPSC-derived ventricular CMs as single cells. Co-immunocytostaining studies performed 7d later demonstrated that $87.3 \pm 0.1\%$ ($n = 24$) of the plated cells expressing cardiac troponin I (cTnI) also expressed MLC-2V.

Interestingly, a small minority of the differentiating cells (~1% and ~5% in the atrial and ventricular differentiation protocols, respectively, in the FACS examples in Fig. 1b, c) were TnT-positive/Nkx2.5-negative. Such TnT$^+$/NKX2.5$^-$ cells, accounting for $5.9 \pm 1.0\%$ ($n = 24$) and $9.1 \pm 1.0\%$ ($n = 35$) of the cells in the ventricular and atrial differentiation protocols, were previously shown to represent sinoatrial node (SAN)-like pacemaker cells[21] and may be responsible for the spontaneous automaticity of the generated atrial and ventricular EHTs.

We next verified that the functional properties of the differentiated cells are consistent with the appropriate ventricular- or atrial-like phenotypes. Patch–clamp analysis of isolated differentiated cells during 1 Hz field stimulation revealed shorter action potentials (APs) in the hESC-derived atrial cells compared with the ventricular cells (Fig. 1d). Consequently, the measured AP duration (APD) at 90%, 50%, and 30% of repolarization were all significantly longer in the ventricular cells compared with atrial cells (APD$_{90}$: $372 \pm 38$ ms vs. $205 \pm 9$ ms, $p < 0.0001$; APD$_{50}$: $310 \pm 33$ ms vs. $147 \pm 10$ ms, $p < 0.0001$; and APD$_{30}$: $266 \pm 29$ ms vs. $115 \pm 9$ ms, $p < 0.0001$; $n = 13$ and 17, respectively, mean ± SEM, unpaired Student's $t$ test, Fig. 1e). The AP maximal upstroke velocity was also steeper in the ventricular cells ($11.8 \pm 1.7$ mV/ms, $n = 13$) when compared with the atrial myocytes ($6.8 \pm 0.8$ mV/ms, $p < 0.01$, $n = 17$, Fig. 1e). In contrast, the average resting membrane potential (RMP) did not differ between the two cells types ($-60 \pm 2$ mV and $-63 \pm 2$ mV for the ventricular and atrial cells, respectively).

Taken together, these results indicate the successful differentiation of hESCs into atrial- and ventricular-like cardiomyocytes that can be used to generate chamber-specific engineered tissues.

### Establishing EHTs.
EHTs were generated by combining the differentiated hESC-derived ventricular or atrial cells with collagen (Fig. 2a). The hydrogel constructs condensed as rings within the casting molds. After initial matrix remodeling (d3), tissues were transferred onto passive silicon stretchers where they started to contract spontaneously (Supplementary Movie 2).

We initially evaluated for potential differences in the gene expression patterns of the chamber-specific EHTs by real-time qPCR (Fig. 2b). The general cardiomyocyte markers *TNNI1*, *TNNT2*, *KCJ2*, and *GJA1* were all expressed in both the atrial and

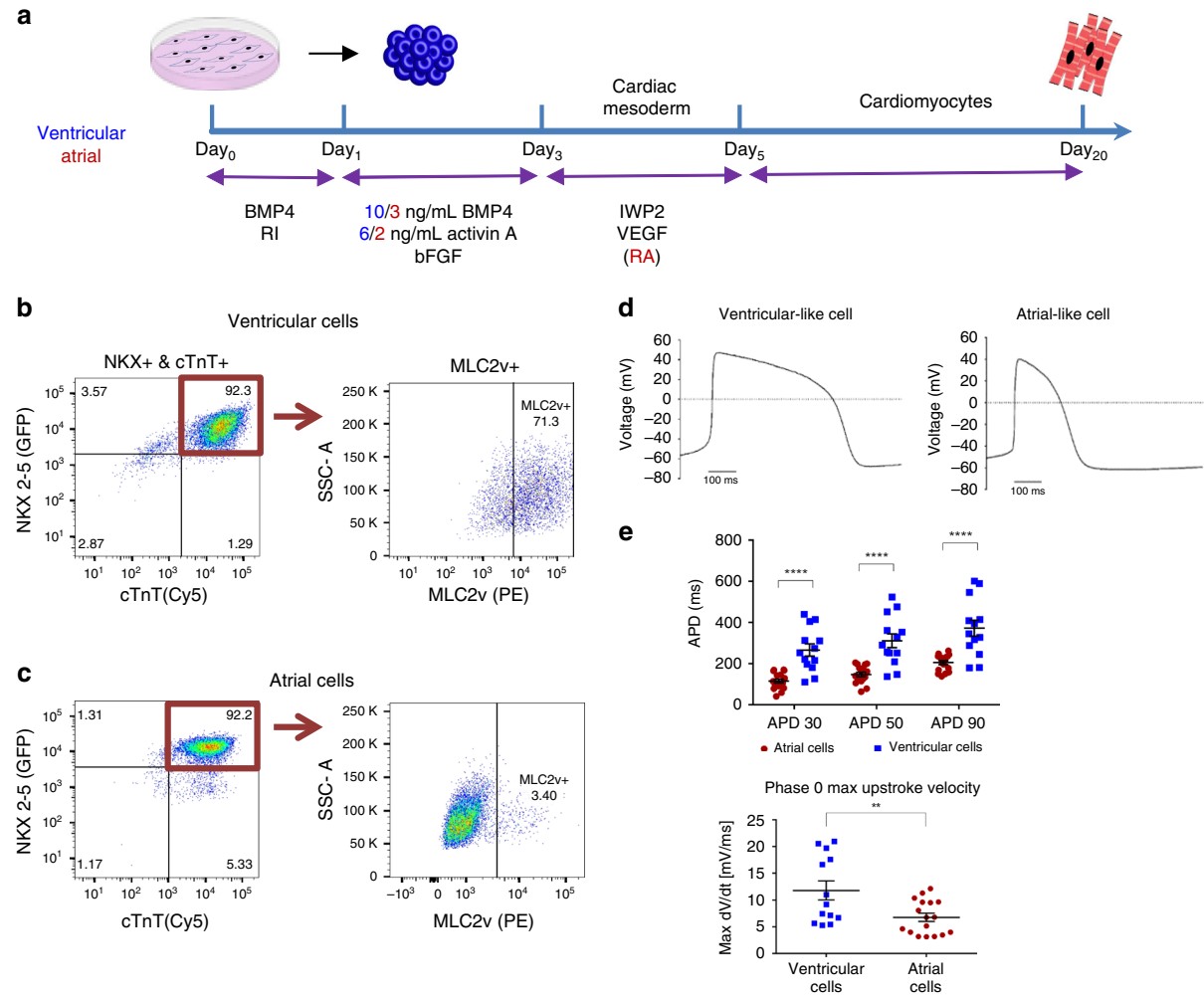

**Fig. 1 Differentiation and characterization of the human embryonic stem cells derived ventricular and atrial cardiomyocytes. a** Schematic representation of the human embryonic stem cells (hESC) differentiation protocols used to derive ventricular (blue) or atrial (red) cardiomyocytes. **b, c** Representative flow-cytometric analyses of the proportion of NKX2.5$^+$/cTnT$^+$ cells (left panels) and the proportion of MLC2V$^+$ cells among these NKX2.5$^+$/cTnT$^+$ cells (right panels) in day 20 HES3-NKX2-5$^{gfp/w}$ EBs using either the ventricular (**b**) or the atrial (**c**) differentiation protocols. Note that using the ventricular and atrial differentiation protocols ~1% and ~5% of the differentiating cells, respectively, were TnT$^+$/Nkx2.5$^-$ cells (sinoatrial node like cells) and that among the differentiating cTnT$^+$/NKX2.5$^+$ cells obtained ~71% and ~3%, respectively, were MLC2v positive. **d** Patch–clamp action potential (AP) recordings of hESC-derived ventricular (left tracing) and atrial (right tracing) cells during 1 Hz field stimulation. **e** Summary of AP duration (APD) measurements from hESC-derived ventricular (blue) and atrial (red) cells (values are expressed as mean ± SEM). Note that hESC-derived atrial cardiomyocytes ($n = 17$, biologically independent cells) displayed shorter APD$_{30}$, APD$_{50}$, and APD$_{90}$ values and decreased phase 0 maximal upstroke velocity compared with the ventricular cells ($n = 13$, biologically independent cells). Recordings were performed at 1 Hz stimulation frequency of. **$p < 0.01$, ****$p < 0.0001$, unpaired $t$ test is used for comparison.

ventricular EHTs (Fig. 2b, top panel). The expression levels of *TNNT2* and *KCJ2* (responsible for the inward rectifier I$_{K1}$ current), which can be used as surrogates for the degree of cardiomyocyte maturity, were similar between the atrial and ventricular EHTs, suggesting a comparable maturation level. Yet, expressions of *TNNT2* and *KCJ2* were lower in both chamber-specific EHTs as compared with their levels in control adult human heart-derived atrial and ventricular tissues (Fig. 2b, top panel).

We next compared the expression levels of genes, known from the literature[1,8,40–43] to be differentially expressed either in atrial (Fig. 2b, middle panels) or ventricular cells (lower panel). These studies revealed significant differences in the expression levels of such chamber-specific genes between the atrial and ventricular EHTs. Thus, the atrial-specific genes *GJA5* (encoding for the gap junction protein connexin 40), *KCNA5* (responsible for the expression of the ultra-rapid potassium current (I$_{Kur}$) in atrial

cells), *KCNJ3* (responsible for the expression of the I$_{KACh}$ potassium current in atrial cells), *NNPA* (encoding for atrial natriuretic factor), *MYL7* (encoding for the myosin regulatory light chain 2, atrial isoform), and *NR2F2* (encoding for the COUP transcription factor 2 known to play an important role in determining atrial identity) were all expressed significantly higher in the atrial EHTs as compared with the ventricular EHTs. These genes were also expressed significantly higher in the control human adult atrial tissue as compared with the control human adult ventricular tissue.

In contrast to the atrial-specific gene expression, the expression levels of the primarily ventricular-specific markers *MYL2* (encoding for the myosin regulatory light chain 2, ventricular isoform), *MYH7* (encoding for the beta-myosin heavy chain), and *HEY2* (a cardiac-specific transcription factor) were significantly higher in the ventricular EHTs as compared with the atrial EHTs. This correlated with their different expression levels in the control

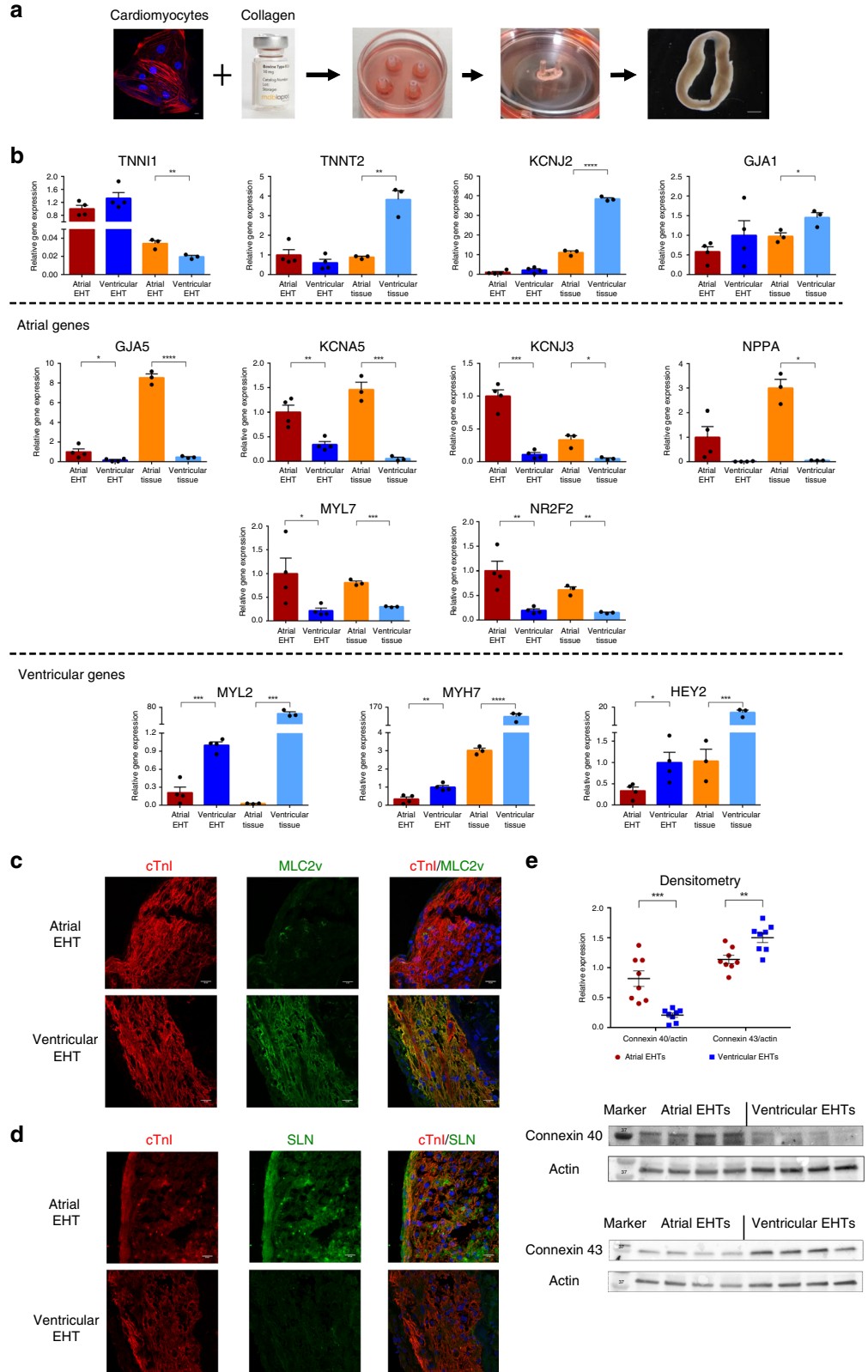

adult human heart-derived atrial and ventricular samples (Fig. 2b, lower panel).

The observed chamber-specific differences between the atrial and ventricular EHTs at mRNA levels were also noted at the protein levels. Thus, co-immunostaining studies targeting the general cardiac sarcomeric protein cTnI and the ventricular-specific marker MLC-2v revealed that the expression of the latter was significantly enriched in the ventricular-EHTs as compared with atrial tissues (Fig. 2c). Morphometric analysis of the stained specimens, quantifying the relative expression of MLC-2v (calculated as the percentage out the total EHT cellular volume that expresses cTnI that also expresses MLC-2v), revealed that it

**Fig. 2 Chamber-specific EHTs generation and characterization. a** Schematic representation of the process of engineered heart tissue (EHT) generation. Chamber-specific human embryonic stem cell-derived cardiomyocytes (hESC-CMs, scale bar: 10 μm) were combined with collagen, solidified in circular molds, and placed on a silicon stretcher for creation of the EHTs (scale bar: 100 μm). **b** Real-time qPCR analysis of the atrial ($n = 4$, biologically independent samples) and ventricular ($n = 4$, biologically independent) EHTs as well as of control adult human atrial and ventricular tissue samples ($n = 3$, biologically independent samples). Shown are the expression levels of the general cardiac-specific markers (TNNI1, TNNT2, KCJ2, and GJA1), atrial-specific markers (GJA5, KCNA5, KCNJ3, NPPA, MYL7, and NR2F2), and ventricular-specific markers (MYL2, MTH7, and HEY2). Expression values were normalized to GAPDH. Error bars represent SEM. *$p < 0.05$, **$p < 0.01$, ***$p < 0.001$, ****$p < 0.0001$; unpaired two-tailed $t$ test. **c, d** Co-immunostaining of 30d atrial and ventricular EHTs for cardiac troponin I (cTnI) and either the ventricular-specific marker MLC2v (**c**) or the atrial marker sarcolipin (SLN) (**d**). Nuclei were stained with DAPI. Scale bars: 20 μm. All eight additional immunostaining images were similar to the representative image shown. **e** Western blot densitometry of Cx40 and Cx43 protein expression in the atrial and ventricular EHTs ($n = 8$, biologically independent samples). All values are expressed as mean ± SEM. **$p < 0.01$, ***$p < 0.001$, Mann–Whitney test is used for comparison.

---

was significantly higher in the ventricular EHTs ($88.8 ± 2.7\%$, $n = 9$) as compared with the atrial tissues ($1.3 ± 0.1\%$, $n = 9$). In contrast to the MLC2v staining, co-immunostaining for cTnI and for the atrial-specific marker sarcolipin (SLN)[20] showed robust expression of the latter protein in the atrial EHTs and its absence in the ventricular tissues (Fig. 2d). Finally, we also evaluated for the expression of non-myocytes cells in the EHTs. Co-immunostaining for the fibroblast marker vimentin and cTnI revealed the presence of a small number of fibroblasts in both atrial and ventricular EHTs (Supplementary Fig. 1a, b).

Another excellent chamber-specific differentiating marker is the type of gap junction protein expressed by the tissue, as connexin 40 (Cx40) is known to be expressed in atrial but not in ventricular cells, whereas connexin 43 (Cx43), the major ventricular gap junction protein, is also expressed in atrial cells. We therefore evaluated in our chamber-specific EHTs, at both the mRNA and protein levels, the expression levels of Cx40 and Cx43. As noted in the qPCR studies, expression of the *GJA5* gene (encoding for Cx40) was significantly higher in the atrial EHTs (Fig. 2b, middle panel). In contrast, *GJA*1 (encoding for Cx43) was expressed in both ventricular and atrial EHTs, with the expression levels being somewhat higher in the ventricular EHTs (Fig. 2b, top panel). We next performed western blot analysis that confirmed the chamber-specific expression pattern of Cx43 and Cx40 also at the protein level (Fig. 2e). Hence, while the atrial EHTs expressed high protein levels of Cx40, its expression in the ventricular EHTs was miniscule ($n = 8$, $p < 0.001$, Mann–Whitney test). In contrast, we noted significant protein expression of Cx43 in both the ventricular and atrial EHTs, with the expression levels being significantly higher in the ventricular EHTs ($n = 8$, $p < 0.01$).

Finally, we also performed transmission electron microscopy of both chamber-specific EHTs, which revealed the presence of cardiomyocytes with organized sarcomeres with a similar degree of ultrastructural maturation (Supplementary Fig. 1c, d). T-tubules, seen abundantly in adult ventricular cells but not in adult atrial myocytes, were absent in both the ventricular and atrial EHTs, indicating their relative early-stage ultrastructural properties.

**Electrophysiological assessment of the atrial and ventricular-engineered tissues**. Atrial and ventricular cardiomyocytes in the human heart differ in their electrophysiological properties due to the expression of a different repertoire of ion channels[43]. Atrial myocytes have shorter APs with distinct triangular-shaped morphology that differ from APs in ventricular cells[43]. To study the electrophysiological properties of the chamber-specific EHTs, we loaded the tissues with the voltage-sensitive dye, FluoVolt, and subjected them to laser-confocal microscopy. We utilized the line-scan mode to allow both high spatial and temporal resolutions, enabling the derivation of optical APs at the cellular resolution within the EHTs during 1 Hz field stimulation. The recorded optical APs displayed distinct morphologies that significantly

differed between the atrial and ventricular EHTs (Fig. 3a). For example, the measured $APD_{90}$, $APD_{50}$, and $APD_{30}$ values were all significantly longer in the ventricular tissues when compared with the respective atrial tissue-constructs ($APD_{90}$: $420 ± 11$ ms vs. $230 ± 5$ ms, $p < 0.0001$; $APD_{50}$: $254 ± 8$ ms vs. $152 ± 4$ ms, $p < 0.0001$; and $APD_{30}$: $190 ± 7$ ms vs. $118 ± 3$ ms; $p < 0.0001$, $n = 42$ and $n = 46$, respectively, in four EHTs each, unpaired Student's $t$ test, Fig. 3b).

To assess the potential use of chamber-specific EHTs for drug testing applications, we initially evaluated drugs known to affect APD. An important distinguishable electrophysiological property between atrial and ventricular cardiomyocytes is the presence of the $IK_{ACh}$ current in atrial, but not ventricular, cells[44]. To evaluate the functional significance of this current in the chamber-specific EHTs, we tested the effects of adding the muscarinic-receptor agonist carbamylcholine at escalating doses (0, 2, 5, and 10 μM); hypothesizing that carbamylcholine will significantly impact only the atrial-EHTs' AP properties. Since carbamylcholine can also change the spontaneous beating rates of the tissues (thereby indirectly affecting APD), we performed these drug studies during continuous 1.5 Hz pacing. Figure 3c shows an example of an optical AP recorded from an atrial-EHT at baseline and the resulting change in AP morphology following application of 10 μM carbamylcholine. Carbamylcholine caused a dose-related shortening of APD in all atrial-EHTs studies (Fig. 3d). For example, 10 μM of the drug significantly prolonged APD from a baseline $APD_{90}$ value of $297 ± 16$ ms to $233 ± 17$ ms ($n = 19$ cells from three independent experiments, $p < 0.01$). In contrast, carbamylcholine had no significant effects on the ventricular EHTs at any dose (Fig. 3d, $n = 19$ cells from three independent experiments, $p = NS$).

Vernakalant is an atrial-selective anti-arrhythmic agent, shown to be highly effective in converting atrial fibrillation (AF)[45] through its effects on several atrial potassium currents[46] to prolong atrial repolarization. Administration of 30 μM of vernakalant significantly prolonged the measured APD values in all atrial-EHTs when compared with baseline values ($APD_{90}$: $248 ± 13$ ms vs. $479 ± 17$ ms, $p < 0.001$; $APD_{50}$: $149 ± 10$ ms vs. $257 ± 16$ ms, $p < 0.001$; and $APD_{30}$: $110 ± 10$ ms vs. $172 ± 13$ ms, $p < 0.01$, unpaired Student's $t$ test; $n = 25$ in three independent EHTs paced at 1.5 Hz; Fig. 3e, left panel). In contrast, vernakalant had no significant effects on the ventricular EHTs' APD (Fig. 3e, right panel, $n = 32$, unpaired Student's $t$ test).

**Optical mapping**. We next proceeded to characterize the electrophysiological properties of the chamber-specific EHTs at the tissue level by analyzing their conduction properties. To this end, the EHTs were loaded with the voltage-sensitive dye Di-4-ANBDQBS and imaged using a high-resolution optical mapping system consisting of a high-speed EM-CCD camera mounted on a fluorescent macroscope. These studies confirmed the development of a functional cardiac syncytium in the chamber-specific

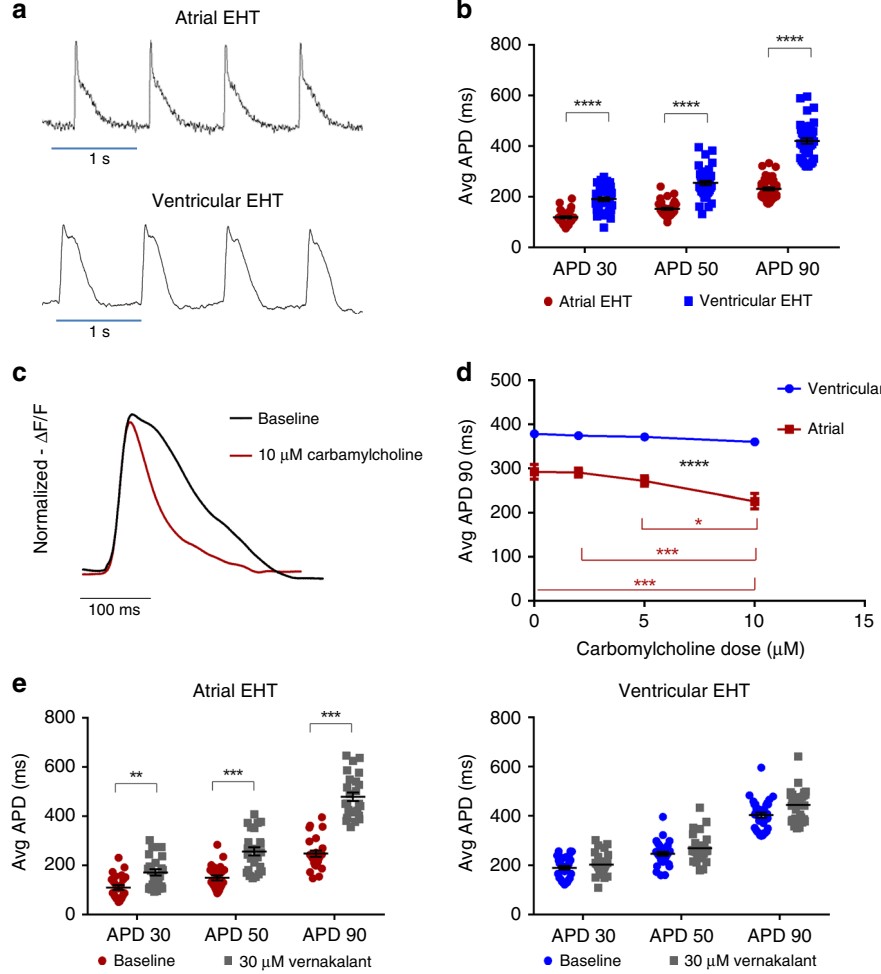

**Fig. 3 Electrophysiological characterization of the chamber-specific EHTs. a** Confocal line-scan images showing examples of optical AP recordings of atrial and ventricular cells within the chamber-specific EHTs. **b** Comparison of mean $APD_{30}$, $APD_{50}$, and $APD_{90}$ values in the atrial and ventricular EHTs ($n = 46$ and $n = 42$ cells, respectively, from three independent experiments) during 1 Hz field stimulation. $****p < 0.0001$, unpaired $t$ test. **c**, **d** Changes in AP morphology (**c**) and duration ($APD_{90}$) (**d**) following application of increasing doses (0, 2, 5, 10 μM) of carbamylcholine and 1.5 Hz field stimulation in the chamber-specific EHTs. Note that significant AP shortening was observed in the atrial ($*p < 0.05$, $***p < 0.001$), but not in the ventricular EHTs ($p = $ NS, repeated-measurements two-way ANOVA followed by Sidak post hoc analysis, $n = 19$ cells from three independent experiments for both the atrial and ventricular experiments). **e** Changes in mean $APD_{30}$, $APD_{50}$, and $APD_{90}$ values following application of 30 μM verankalant during 1.5 Hz field stimulation. The atrial cells ($n = 25$ cells from three independent experiments) showed significant prolongations while the effect on the ventricular cells ($n = 32$ cells from three independent experiments) was not statistically significant. All values are expressed as mean ± SEM. $**p < 0.01$, $***p < 0.001$, unpaired $t$ test.

EHTs, with synchronized AP wave propagation (Supplementary Movie 3).

Interestingly, a significant difference was noted in the baseline rhythm of the ventricular and atrial EHTs. In the ventricular tissues, we observed a normal activation pattern that was initiated at a primary pacemaker region developing in each EHT, with the activation wavefront then propagating to synchronously activate the rest of the ventricular-EHT (Fig. 4a; Supplementary Movie 3). In contrast, the majority of the atrial-EHTs displayed different types of arrhythmias at baseline. These rhythm disorders were all re-entrant in nature, but varied in their level of complexity (Fig. 4b; Supplementary Movie 4). In some cases, we detected a large single circular re-entry wave propagating around the generated ring (Fig. 4b, left panel; Supplementary Movie 4). In other atrial-EHTs, we noted the presence of a single or multiple spiral-wave reentrant loops (Fig. 4b, middle and right panels; Supplementary Movie 4). The biophysical properties of these rotors could be analyzed through the use of both activation (Fig. 4b) and phase maps (Fig. 4c).

In most of the arrhythmic atrial-EHTs, we were able to terminate the arrhythmias by applying an electrical field potential (electrical shock) using a prolonged field stimulation pulse. This resulted in arrhythmia conversion into normal activity with synchronous AP propagation from a single pacemaker region (Fig. 4d; Supplementary Movie 5). These results highlight the potential of the atrial-EHT as a model for atrial arrhythmias, such as AF.

After resumption of normal rhythm in the atrial engineered tissues, we studied and compared the conduction properties of the atrial and ventricular EHTs under similar conditions using continuous point electrical stimulation (Fig. 4e). To this end, we paced the chamber-specific EHTs at increasing frequency (1.5–3.5 Hz) and analyzed the resulting optical activation maps to measure the conduction velocity (CV) values. Significantly faster CVs were observed in the ventricular-EHTs as compared with the atrial tissues at all pacing frequencies, as can be appreciated in the corresponding CV versus pacing cycle length (CL) restitution plots (Fig. 4f, left panel). For example, the

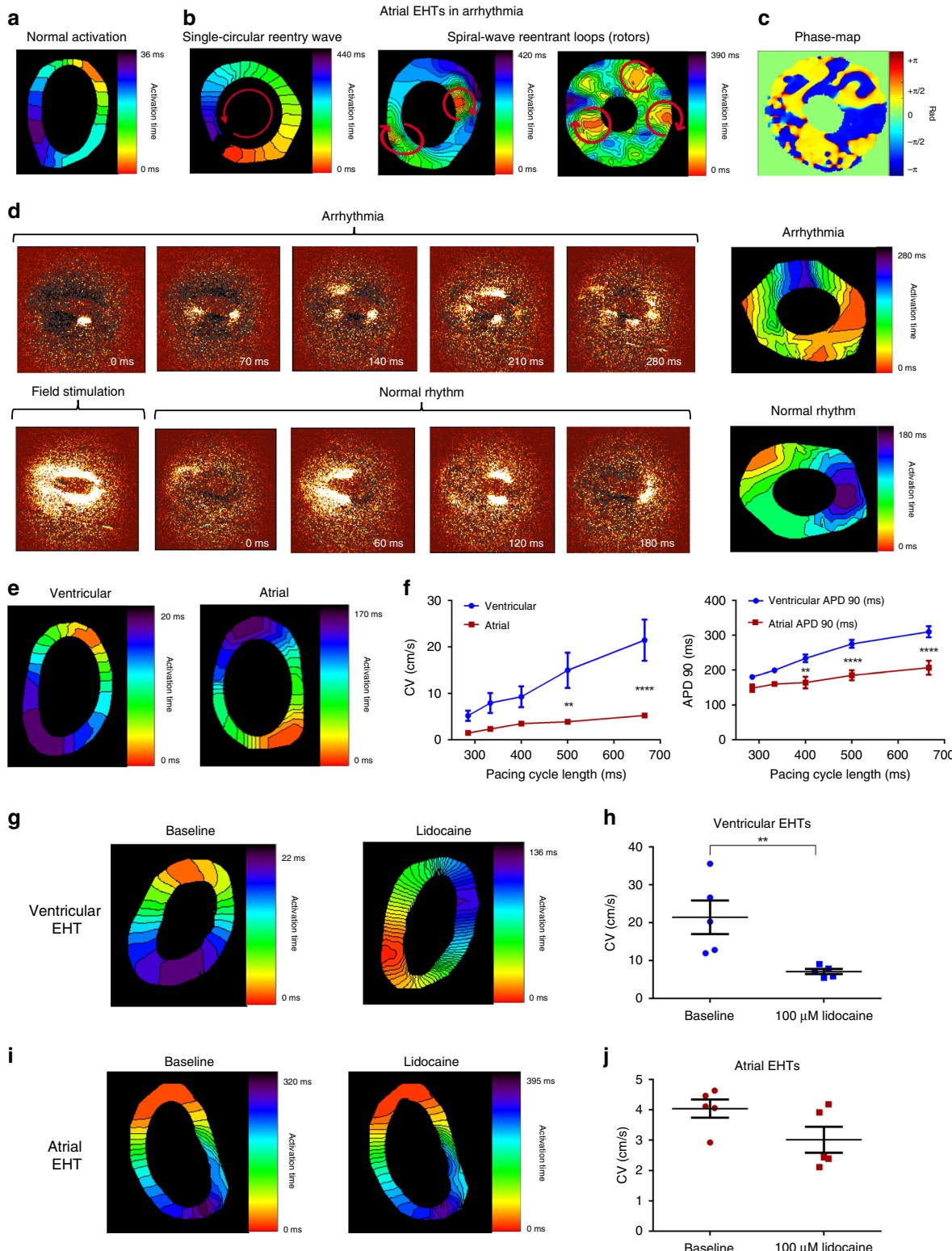

average CV measured at a pacing frequency of 1.5 Hz was significantly higher in the ventricular EHTs ($21.4 \pm 4.4$ cm/s, $n = 5$, $p < 0.0001$, repeated-measurements two-way ANOVA followed by Sidak post hoc analysis) versus the atrial tissues ($4.1 \pm 0.2$ cm/s, $n = 5$). Similar to the CV restitution plots, we could also derive $APD_{90}$ restitution curves, which demonstrated APD shortening at faster pacing rates (shorter CLs). Similar to the single-cell experiments, APD values were significantly shorter in the atrial

EHTs when compared with the ventricular tissues at all pacing CLs studied ($n = 6$, Fig. 4f, right panel).

To examine the effective refractory period (ERP) of the atrial and ventricular EHTs, the tissues were paced with a train of ten stimulated APs at a fixed CL (S1), followed by a premature stimulus (S2). ERP was measured by gradually shortening the coupling interval of the premature stimulus (S1–S2) until loss-of-capture (Supplementary Fig. 2a). These studies revealed that the

**Fig. 4 Optical mapping of the chamber-specific EHTs. a** Activation map of a ventricular EHT during spontaneous activation showing the propagation of the electrical wavefront throughout the EHT from the earliest (red) to latest (purple) activation sites. **b** Activation maps of atrial engineered tissue during spontaneously-occurring different re-entrant arrhythmias including: a single anatomical macro-reentrant loop (left panel) and multiple spiral waves (middle and right panels). **c** A snap shot of a phase map characterizing the rotors in the atrial EHT. Colors represent phase as indicated in the color bar. **d** Time-lapse sequences (left panels) and the resulting activation maps (right panels) of wavefront propagation in an atrial EHT during an arrhythmia, during application of field stimulation, and following the renewal of normal activity. **e** Activation maps of ventricular and atrial EHTs during point stimulation at a frequency of 1.5 Hz. **f** Changes in conduction velocity (CV) values (left panel, $n = 5$, biologically independent samples) and $APD_{90}$ (right panel, $n = 6$, biologically independent samples) in the atrial and ventricular EHTs as function of the pacing cycle length (**$p < 0.01$, ****$p < 0.0001$, repeated-measurements two-way ANOVA followed by Sidak post hoc analysis). **g–j** Effects of addition of 100 μM lidocaine on the resulting activation maps (**g**, **i**) and mean CV values (**h**, **j**) of ventricular (**g**, **h**) and atrial (**i**, **j**) EHTs ($n = 5$ biologically independent samples). Recordings were performed at a stimulation frequency of 1.5 Hz. Values are expressed as mean ± SEM. **$p < 0.01$, Mann–Whitney test.

ventricular EHTs were characterized by significantly longer ERP values ($240.0 \pm 9.4$ ms, $n = 10$) in comparison to the atrial EHTs ($86.0 \pm 2.9$ ms, $n = 10$, Supplementary Fig. 2a, b).

We next studied the effect of lidocaine, a sodium channel blocker used for treating ventricular (but not atrial) arrhythmias, on the conduction properties of the chamber-specific EHTs. The EHTs were studied during pacing (1.5 Hz) at baseline and following application of 100 μM lidocaine. As shown in the resulting activation maps (Fig. 4g), the addition of lidocaine to the ventricular EHTs significantly slowed conduction (~66% decrease in mean CV values, Fig. 4h, $n = 5$, $p < 0.05$, Mann–Whitney test). In contrast, the effect of lidocaine on slowing CV in the atrial-EHTs was much milder and did not reach statistical significance (Fig. 4i, j).

Finally, we evaluated the usefulness of our established atrial EHT-based arrhythmia (AF-like) model for pharmacological testing. To this end, we evaluated the ability to convert the atrial arrhythmias into normal rhythm by exposing the arrhythmogenic atrial-EHTs to the anti-arrhythmic agents, vernakalant and flecainide (pharmacological cardioversion) as well as to field stimulation (electrical cardioversion). As can be seen in Supplementary Movie 6 and Fig. 5a, b, both flecainide (10 μM) and verankalant (30 μM) converted the arrhythmic EHTs into normal rhythm after 10 min of treatment. Field stimulation was also successful in restoring normal rhythm (Supplementary Movie 5). The success rate of converting the arrhythmias into normal rhythm was highest for field stimulation (86.6%, $n = 30$), followed by 10 μM flecainide (77.7%, $n = 9$), and 30 μM vernakalant (52.9%, $n = 17$) (Fig. 5c).

Interestingly, the vast majority (84.6%, $n = 26$, Fig. 5d) of the atrial-EHTs that were converted to a regular rhythm by field stimulation resumed arrhythmogenic activity spontaneously within 15 min. The patterns of the reentrant arrhythmias observed in these recurrences varied and differed from the original arrhythmia pattern in 40.9% of the cases. In contrast, most of the atrial-EHTs that were converted to regular rhythm and then continuously treated with anti-arrhythmic agents remained in normal rhythm (100% and 66.6% of EHTs treated with flecainide and verankalant, respectively, Fig. 5d). In the few cases where arrhythmias did re-occur their pattern was similar to the original reentry type.

**Mechanical force measurements**. In addition to the electrophysiological characterization of the tissues, we also evaluated their contractile properties. To this end, we utilized an experimental setup consisting of a sensitive force-transducer coupled with a length controller to study the mechanical properties of the tissues (Supplementary Fig. 3a). This allows direct measurement of both the passive and active forces of the atrial and ventricular EHTs. The constructs were then subjected to a series of length increases while being paced at a constant frequency (Fig. 6a). Forces were continuously evaluated during the stretching of the

constructs allowing force measurement at each resting length (Fig. 6b).

The active force generated was calculated as the amplitude of the recorded twitch force at each specific tissue-construct length (Fig. 6c). We then normalized the forces measured per cross-sectional area (CSA), and plotted the active force against the changes in construct length. In all tissues, regardless of cardiac subtype, increments in tissue length enhanced active force production (Fig. 6d). Passive forces rose as well with tissue stretching in both EHT types (Supplementary Fig. 3b). The increase in measured forces with tissue stretching confirmed that both ventricular and atrial EHTs fulfill the fundamental force–length (Frank-Starling) relationship of cardiac muscle.

Comparison of the active force developed by the ventricular and atrial EHTs showed that the ventricular-engineered tissues developed significantly higher active forces than the atrial specimens at all tissues' lengths (Fig. 6d). For example, at the initial resting state, the mean active force generated by the ventricular-EHTs ($0.92 \pm 0.09$ mN/mm$^2$, $n = 12$) was significantly higher than that produced in the atrial-EHTs ($0.19 \pm 0.04$ mN/mm$^2$, $n = 12$ $p < 0.0001$, Mann–Whitney test, Fig. 6e).

Finally, we studied the contraction forces of the atrial and ventricular EHTs under the effect of different treatments. Increasing extracellular calcium concentration (0.2–3.2 mM) resulted in enhanced inotropic responses of both tissues types (Fig. 6f, g). Interestingly, the nature of this calcium dependency differed between the atrial and ventricular EHTs, with the atrial EHTs requiring a higher calcium concertation to reach 50% of the maximal force (Fig. 6f, g).

We also studied the effect of drugs with opposing ionotropic actions on the EHTs. These studies were performed at the resting tissue-construct tension and during 2 Hz of constant field stimulation. Application of 10 μM isoproterenol, a β-adrenergic agonist, to both chamber-specific tissues (Fig. 6h; Supplementary Fig. 4a) significantly increased their forces of contraction by >70% (Fig. 6i). In contrast, incubation with the negative inotropic agent L-type calcium channel blocker nifedipine (0.1 μM) significantly reduced the force of contraction in both the ventricular (Fig. 6j) and atrial (Supplementary Fig. 4b) EHTs by >50% (Fig. 6k).

## Discussion
The ability to combine hPSCs technologies, advanced cardiomyocyte differentiation protocols, and various tissue engineering strategies facilitated the establishment of engineered human cardiac muscle tissue-constructs, with important implications for cardiovascular regenerative medicine, disease modeling, and drug discovery[28–39]. Until now, however, such engineered tissues were assembled mostly from a mixture of different cardiomyocyte cell types (ventricular-, atrial-, and nodal-like cells), which presented a major challenge in fulfilling their potential for the aforementioned applications. In this study, similar to other attempts in the field[47,48], we aimed to address this challenge by utilizing a

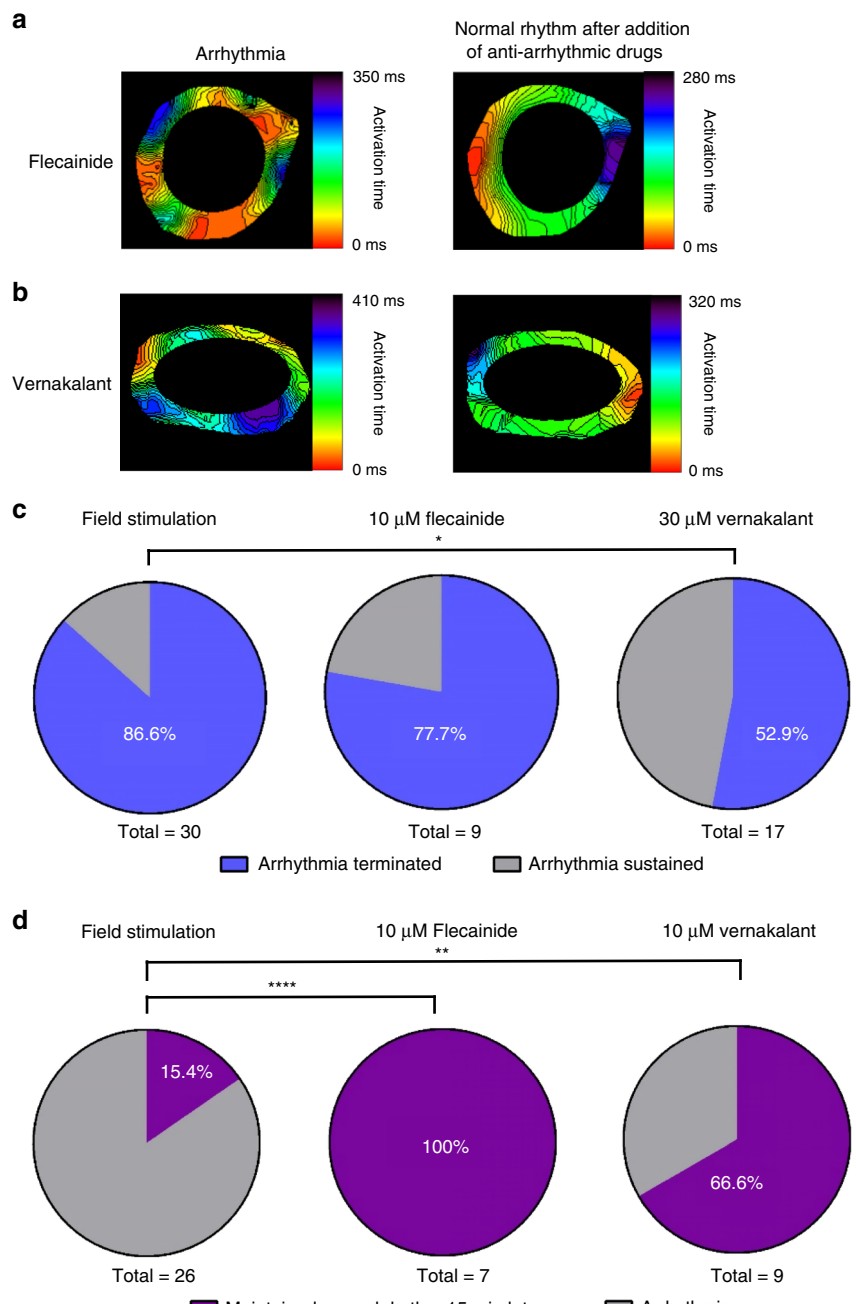

**Fig. 5 Electrical and pharmacological cardioversion of atrial arrhythmias. a, b** Activation maps of the arrhythmogenic atrial EHTs, during arrhythmia and after conversion to normal rhythm by 10 μM flecainide (**a**) or 30 μM vernakalant (**b**) treatments. **c** Percentage of successful arrhythmia termination in the atrial EHTs following field stimulation ($n = 30$, biologically independent samples) or the addition of 10 μM flecainide ($n = 9$, biologically independent samples) or 30 μM verankalant ($n = 17$, biologically independent samples). *$p < 0.05$, Fisher test. **d** Success rate (%) in maintaining normal rhythm at 15 min after cardioversion without any treatment ($n = 26$, biologically independent samples), with flecainide treatment ($n = 7$, biologically independent samples) and with verankalant treatment ($n = 9$, biologically independent samples). **$p < 0.01$, ****$p < 0.0001$, Fisher test.

development biology-guided hPSCs differentiation strategy[8] to generate chamber-specific cardiomyocytes and eventually to derive ventricular and atrial EHTs.

Our results demonstrate: (1) the ability to robustly generate both ventricular and atrial EHTs, using a collagen-based hydrogel approach; (2) that the atrial and ventricular EHTs differentially express several chamber-specific markers (including unique transcription factors, sarcomeric, ion channel, and gap junction protein subtypes) at both the RNA and protein levels; (3) that the generated chamber-specific EHTs display distinct ventricular or atrial molecular, electrophysiological, and contractile properties;

(4) the ability to use this strategy for drug testing by demonstrating chamber-specific functional effects following application of atrial/ventricular-specific pharmacology; (5) the ability to study complex electrophysiological phenomena such as conduction and reentrant arrhythmias and specifically the potential of using the atrial-EHT as a model to study AF; and (6) the ability to characterize the mechanical properties of the chamber-specific EHTs, with the ventricular tissues producing significantly greater contractile forces than the atrial-EHTs.

Examining the electrophysiological properties of the chamber-specific EHTs revealed triangular-shaped and shorter optical APs

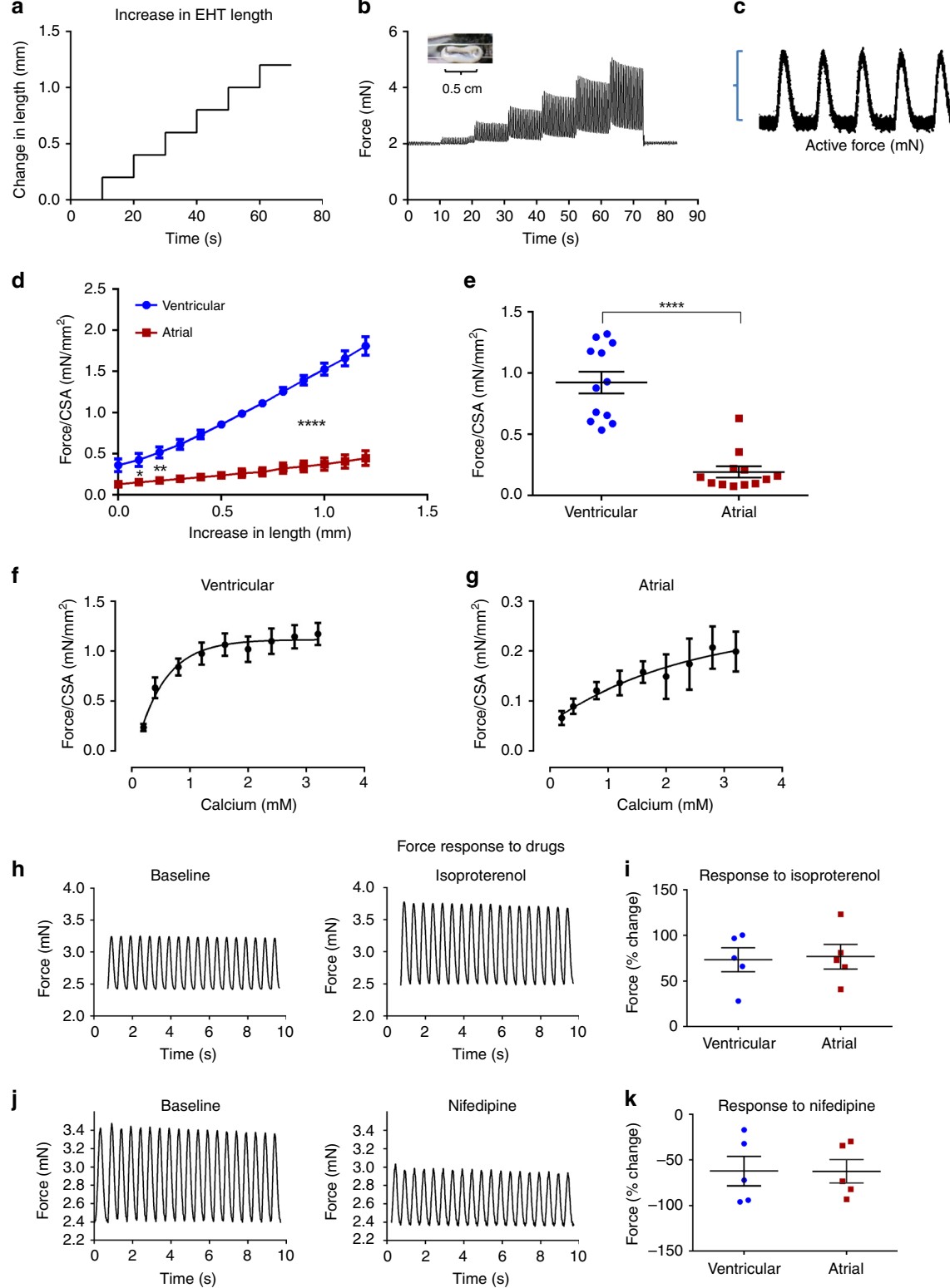

in the atrial-EHTs compared with ventricular tissues, in correlation to what is known of the human heart[43]. We next examined the effect of drugs known to display differential activity on atrial and ventricular tissues. Carbamylcholine, known to open the atrial $I_{KACh}$ channel[49], had a dose-related effect to significantly shorten the APD in the atrial-EHTs, whereas its effect on the ventricular-EHTs was negligible. Another agent known to specifically affect atrial electrophysiology is vernakalant[45,46]. This anti-arrhythmic

agent, which affects a variety of sodium and potassium channels, displays a marked specificity for blocking the atrial-specific currents $I_{Kur}$ and $I_{KACh}$, thereby specifically prolonging of the atrial APD and refractory period[46]. Consistent with these known actions, vernakalant significantly prolonged APD in our atrial-EHTs, whereas it had no significant effect on the ventricular tissues.

To study more complex electrophysiological phenomena such as conduction and re-entrant arrhythmias, we combined our

**Fig. 6 Mechanical force assessment in the chamber-specific EHTs. a** A series of length increases of the EHTs is controlled by a highly sensitive motor. **b**, **c** Tracings depicting the measured EHT forces over time during sequential EHT stretching and field stimulation (**b**) and a high-magnitude example of the measured active force (**c**). **d** The active forces were measured and normalized to cross-sectional area (CSA), and then plotted as function of EHT length for both ventricular (blue) and atrial (red) EHTs. Note the typical length–tension relationships with the generated forces increasing for both tissue types as function of tissue length ($n = 6$, biologically independent samples). Also notice the significantly higher forces developing in the ventricular EHTs as compared with the atrial tissues. *$p < 0.05$, **$p < 0.01$, ****$p < 0.0001$, repeated-measurements two-way ANOVA followed by Sidak post hoc analysis. **e** Active forces developed by the ventricular and atrial EHTs with similar initial lengths (stimulated at 2 Hz and normalized per CSA). Note that the ventricular tissues developed significantly higher mean active force than the atrial tissues ($n = 12$, biologically independent samples). ****$p < 0.0001$, Mann–Whitney test. **f**, **g** Changes in contractile forces (normalized per CSA) in response to increasing calcium concentration for the ventricular (**f**) and atrial (**g**) EHTs ($n = 4$, biologically independent samples, for both EHT types). **h-k** Changes in contractile forces of the ventricular EHT following addition of 10 µM of isoproterenol (**h**, **i**) and 0.1 µM of nifedipine (**j**, **k**). Shown are the actual force tracings (**h**, **j**) and summary of the percentage change (**i**, **k**) in the mean measured forces of the atrial (red) and ventricular (blue) tissues in response to 10 µM of isoproterenol ($n = 5$, biologically independent samples, **i**) or 0.1 µM nifedipine ($n = 5$, biologically independent samples, **k**). All recordings were done at a stimulation frequency of 2 Hz. Values are expressed as mean ± SEM.

chamber-specific EHT models with a high-resolution optical mapping system, previously established in our laboratory to study 2D cardiomyocyte monolayers[50]. These studies revealed the development of a functional cardiac syncytium in both types of EHTs, with synchronized AP propagation. Under identical pacing stimulation, CV in the ventricular EHTs was faster than in the atrial tissues. One potential mechanism underlying this difference may be the steeper upstroke velocity of the ventricular APs (compared with the atrial APs) as noted in the patch–clamp studies. Another potential contributing factor may be the type of gap junctions protein expressed by the chamber-specific EHTs. Hence, while the ventricular EHTs, similar to the adult ventricle, express high levels of Cx43 (thereby assembling primarily homotypic Cx43-Cx43 gap junctions); the atrial EHTs, in a similar manner to the in vivo atria, express both Cx40 and to a lesser extent Cx43 (thereby potentially also assembling homotypic and heteromeric gap junctions consisting of a mixture of these proteins). This difference may impact the tissue's impulse propagation, as shown elegantly by Beauchamp et al.[51], who studied atrial tissue strands prepared from a combination of control and homozygous or heterozygous mice with genetic deficiency of either Cx43 or Cx40. Their results demonstrated that dominance of Cx40 decreases electrical wave propagation velocity whereas dominance of Cx43 increases the speed of conduction.

Interestingly, CVs measured in our chamber-specific EHTs (and specifically in the ventricular EHTs) were significantly higher than most previous reports evaluating hPSC-derived 2D monolayer-based or 3D EB-based models[50,52,53] and were similar to other EHT models[27,28] indicating a relative high degree of tissue structural organization and electrophysiological maturation. Consequently, the CV values measured in the ventricular EHT model (>20 cm/s) are now in the same magnitude, albeit still somewhat lower, as values reported in the in vivo human atria (~50 cm/s)[54,55] and ventricles (50–100 cm/s)[56]. Finally, we also showed the utility of our chamber-specific tissue models for evaluating drug effects on conduction. Application of the class Ib anti-arrhythmic agent lidocaine, for example, which is used to treat ventricular (but not atrial) arrhythmias, significantly slowed CV in the ventricular-EHTs while having limited (and not significant) effects on the atrial-EHTs.

An important implication of this study is the ability to utilize the chamber-specific EHT models to study re-entrant arrhythmias, and specifically to focus on atrial arrhythmias such as AF. Interestingly, re-entrant arrhythmias developed spontaneously in most atrial-EHTs. In contrast, the ventricular EHTs seldom developed arrhythmia spontaneously or following electrical stimulation protocols. This difference in arrhythmias susceptibility probably stems from the marked electrophysiological differences observed between the two tissue types. Since the tissue

wavelength (WL) is a product of the tissue refractory period and CV, the slower CV and shorter refractory period (or its surrogate measurement-APD) observed in the atrial-EHTs resulted in a markedly shorter WL in this tissue type as compared with the ventricular-EHTs. Since a shorter tissue WL is associated with a greater probability for initiation and maintenance of re-entrant circuits, this explains the marked differences in arrhythmogenicity between the two EHT types. This finding may also explain why AF and other atrial arrhythmias are more common clinically and are easier to induce than ventricular arrhythmias, despite the larger ventricular tissue mass.

The development of the atrial-EHT arrhythmia model also allowed us to evaluate the effects of various therapeutic interventions. Consequentially, we were able to interrupt/terminate arrhythmic activity in the atrial-EHTs by clinically relevant interventions, such as electrical and pharmacological cardioversion. Electrical field stimulation was the most effective modality with ~90% arrhythmia termination rates, with pharmacological treatments (flecainide and verankalant) also being highly effective. Moreover, the atrial-EHT model was useful as a drug-testing platform, both for studying drugs' efficacy in terminating arrhythmias (pharmacological cardioversion) and in preventing initiation of new arrhythmias (rhythm-control strategy).

As a proof-of-concept for demonstrating the utility of our atrial-EHTs, we studied the effect of the well-established type Ic anti-arrhythmic agent flecainide and showed its ability to terminate the atrial arrhythmias in the atrial-EHT model. To demonstrate the power of our system to predict the effects of relatively novel drugs, we also evaluated the effects of verankalant, which was recently introduced into clinical practice because of its high success rate in converting acute AF[45]. Verankalant terminated the re-entrant activity in our atrial-EHT model and, to a lesser extent, also prevented the development of new arrhythmias. The chamber-specific EHT models also revealed the relative selective action of vernakalant in primarily affecting the atrial (prolonging APD), but not the ventricular, electrophysiological properties. It is thought that the effects of verankalant result from its action on multiple currents[46,57]. Vernakalant is known to induce a frequency- and voltage-dependent $I_{Na}$ block, which is probably its most important action with regard to AF termination since it preferably effects intra-atrial conduction at fast rates. In addition, the drug also inhibits the early activating K$^+$ channels ($I_{Kur}$; $I_{to}$) and $I_{KACh}$[57]. Since, $I_{Kur}$ and $I_{KACh}$ are atrial-specific currents, this also contributes to the relative atrial selectivity of the drug in prolonging refractory period and its efficacy in terminating AF[57].

Finally, we characterized and compared the mechanical properties of the ventricular and atrial EHTs. In both chamber-specific tissues, we observed the typical length–tension relationships

(Frank-Starling mechanism) of cardiac muscle[58]. The forces recorded in the EHTs (~2 mN/mm) are in a similar range to other previously reported tissue engineered cardiac muscle models utilizing hPSC-CMs, with some studies reporting higher values[28,59] and others lower or similar force magnitude[60,61]. The forces measured in both chamber-specific EHTs, however, were significantly lower than those generated by the adult human heart muscle[62–65]. Interestingly, the ventricular-EHTs generated higher forces of contraction compared to the atrial tissues, correlating with previous findings in atrial and ventricular fibers derived from healthy human hearts[65,66]. Both chamber-specific EHTs displayed increased forces of contraction following elevation of extracellular $Ca^{2+}$ levels, with the atrial-EHTs requiring higher $Ca^{2+}$ concentration to reach 50% of the maximal contraction amplitude. This is consistent with early studies showing that human atrial tissue is less sensitive to calcium than ventricular tissues[65,67]. Also, both chamber-specific EHTs responded to pharmacological agents known to affect contractility, with isoproterenol and nifedipine causing an increase and decrease in the developed forces, respectively.

The chamber-specific EHT models presented here also possess certain limitations. The hPSC-CMs and the resulting chamber-specific EHTs are not as mature as the adult human cardiac tissue, as evident, for example, by the reduced expression levels of maturation-related genes such as *TNNT2* and *KCJ2* (responsible for the inward rectifier $I_{K1}$ current) and by the lack of t-tubules. Moreover, despite improvements relative to previously described hPSC-derived tissue models, some important functional parameters such as the cells' resting membrane potential (which is relatively depolarized at ~−60 mv), the measured conduction velocities, and the amplitudes of the mechanical forces generated still did not reach the same level of maturity as the adult heart. This early maturity state represents a general limitation in the hPSC-CMs field[68], and similar results were observed in other hPSC-based EHT models[22,61]. This limitation may be ameliorated by ongoing efforts in the field to generate a more mature hPSC-CMs by using different hormonal treatments[69,70], optimizing extracellular matrix composition[71], mechanical and electrical training[28], and a variety of other interventions[68]. Finally, the macroscopic structure of the generated tissues was relative simple (ring-shaped EHTs) and did not differ between the atrial and ventricular tissues. Ongoing and future efforts using either decellularized organs or advanced three-dimensional bio-printing[48] will attempt to also combine the atrial/ventricular cells with the relevant chamber-specific structures.

In conclusion, despite the aforementioned limitations, our results show that chamber-specific hPSC-based EHTs can be generated, and that the derived atrial- and ventricular-engineered heart tissues recapitulate fundamental structural and functional properties ascribed to each cardiac chamber. These models can be used in several cardiac research fields, such as developmental biology, disease modeling (including of AF), drug development, and drug testing.

## Methods

**Differentiation into ventricular and atrial cardiomyocytes.** The HES3-NKX2-5$^{egfp/w}$ reporter hESC line (kind gift from Prof. Elefanti) was used in this study. The approach for ventricular and atrial differentiation was recently described[8], and is outlined in Fig. 1a. Briefly, undifferentiated hESC colonies were grown on MEF feeder layer, and maintained with the DMEM/F12 medium supplemented with 1% penicillin/streptomycin, 2 mM L-glutamine, 1× nonessential amino acids, 55 μM β-mercaptoethanol, and 20% KnockOutTM serum replacement. To induce differentiation, cells were cultured as embryoid bodies (EBs) and cultivated in suspension in backbone medium containing StemPro-34 (Thermo-Fisher) supplemented with 1% penicillin/streptomycin, 2 mM glutamine, 150 μg/ml transferrin (Roche), 50 μg/ml ascorbic acid, and 50 μg/ml monothioglycerol (MTG). Culture medium was supplemented on day 0 (for 18 h) with 1 ng/ml rhBMP4 (R&D) and 10 μM ROCK inhibitor Y-27632 (TOCRIS). On day 1, backbone

medium was supplemented with rhBMP4 (10 ng/mL for ventricular or 3 ng/mL for atrial differentiation), rhActivinA (10 ng/mL or 2 ng/m for ventricular/atrial differentiation, R&D) and 5 ng/ml rhbFGF. On day 3, medium was supplemented with 10 ng/mL rhVEGF (R&D), 1 μM IWP2 (Wnt inhibitor, TOCRIS), and for atrial differentiation also with retinoic-acid (RA, 0.25–0.5 μM, Sigma). On day 5, culture medium was replaced with CDM3 medium[72] composed of RPMI-1640, 0.33% l-ascorbic acid 2-phosphate (AA 2-P), 0.66% human albumin, and 1% penicillin/streptomycin. After 20 days of culture, EBs were enzymatically dissociated into single cells using collagenase2 (Worthington) and TrypLE (Gibco) for flow cytometry, patch–clamp analysis, immunocytostaining, and creation of engineered tissues.

**Flow cytometry.** For intracellular staining of cardiac troponin T (cTnT) and myosin light chain 2 ventricle (MLC-2V), dissociated cardiomyocytes were fixed with 4% paraformaldehyde, permeabilized in PBS with 5% FCS and 0.5% Saponin, and washed again with PBS containing 0.5% BSA (Sigma). Cells were then stained overnight (4 °C) with the unconjugated primary antibodies targeting cTnT (MA5-12960, Thermo-Fisher Scientific) and MLC-2V (ab79935, Abcam) in FACS buffer consisting of PBS with 5% fetal bovine serum (FBS, Gibco). Cells were then washed again and stained with secondary antibodies (diluted in PBS with 5% FCS) for 1 h at 4 °C. Finally, cells were evaluated with the LSR FortessaII flow cytometer (BD-Biosciences), and FlowJo software.

**Patch–clamp studies.** The action potentials (APs) of ventricular and atrial single cells were measured in whole-cell patch clamp configuration. Cells were plated on glass coverslips, and APs were recorded from spontaneously contracting cardiomyocytes kept in Tyrode's solution (NaCl 140 mM, KCl 5.4 mM, CaCl$_2$ 1.8 mM, MgCl$_2$ 1 mM, HEPES 10 mM, and glucose 10 mM, pH, 7.4). The pipette solution consisted of KCl 120 mM, MgCl$_2$ 1 mM, Mg-ATP 3 mM, HEPES 10 mM, and EGTA 10 mM, pH 7.2. APs were recorded and analyzed using the current-clamp mode using Axopatch 700B, Digidata 1440 A, and pClamp 10 (Axon Instruments). APs were recorded during 1 Hz stimulation.

**Generation of the engineered heart tissues.** Engineered heart tissues (EHTs) were generated by combining 2 million hESC-derived atrial/ventricular cells with bovine collagen (LLC Collagen-Solutions) and 2× DMEM (containing 40% 5× DMEM, 40% fetal bovine serum, 15% H$_2$O, 10 μl/ml glutamine, and 20 μl/ml penicillin/streptomycin). The mixture was pipetted into circular casting molds, where it condensed for 3 days and then transferred onto a silicon passive stretcher. Medium (Iscove-Medium with 20% fetal bovine serum, 1% nonessential amino acids, 1% glutamine, 1% penicillin/streptomycin, and 100 μmol/l β-mercaptoethanol) was changed every other day.

**Real-time quantitative PCR.** The total RNA was isolated using RNeasy Fibrous Tissue Mini Kit (QIAGEN; 74704) according to the manufacturer's protocol. DNAse treatment was performed on 500 ng RNA using RQ1 RNAse-free DNAse. First strand cDNA was generated from 100 ng RNA of each sample using SuperScript II Reverse Transcriptase. qPCR reactions were performed in triplicates using LightCycler 480 SYBR Green I Master and using the LightCycler 480 instrument. mRNA relative levels were calculated based on the comparative threshold cycle (Ct) method. The Ct for each mRNA and endogenous control GAPDH in each sample were used to create ΔCt values [Ct(mRNA) – Ct(GAPDH)]. For cardiac and atrial genes, relative quantification (RQ) was calculated by subtracting the average ΔCt of ventricular samples from the average ΔCt of atrial samples, and using the equation: RQ = $2^{-\Delta\Delta Ct}$. The calculation for ventricular genes was done in a similar way, using the average ΔCt of ventricular samples as a reference. Primer sequences are listed in Supplementary Table 1.

**Immunostaining.** EHTs were frozen and cut into sections (10 μm). The slides were permeabilized with 1% Triton X-100 and then blocked with 5% horse serum. They were subsequently incubated with primary antibodies cTnI (1:150, MAB 1691, Merck), MLC-2v (1:150, ab79935, Abcam), SLN (1:150, sc-46261, Santa Cruz) and vimentin (1:150, ab92547, Abcam) at 4 °C overnight. Specimens were washed and stained with appropriate secondary antibodies Cy2, Cy3, Cy5 (1:150, Jackson) and DAPI. The preparations were examined using a Zeiss LSM-710 laser-scanning confocal microscope. For quantitative volumetric analysis of the images obtained, the Imaris software was used.

To quantify the percentage of ventricular cells obtained using the ventricular differentiation protocol, differentiating hPSC-derived ventricular CMs were plated as single cells. After 7d in culture, we performed co-immunostaining studies of the plated cells for cTnI and MLC-2v. The resulting images were analyzed using the Imaris software.

**Transmission electron microscopy.** The tissue samples were fixed in 3.5% glutaraldehyde and 0.1 M sodium cacodylate buffer (pH 7.4). Blocks of 1 mm$^3$ were prepared and post-fixed (1 h) with 2% OsO4 in 0.2 M cacodylate buffer. To remove excess osmium, samples were rinsed again in cacodylate buffer. Tissue blocks were then immersed in saturated aqueous uranyl acetate, dehydrated in graded alcohol

solutions, immersed in propylene oxide, and finally embedded in Epon 812. Sections (80 nm), derived from the blocks, were attached to a thin-bar copper grid, and counterstained with saturated uranyl acetate and lead citrate. Examination of the different sections was performed with the Jeol1011-JEM electron microscope at 80 KV.

**Western blot analysis**. EHTs were lysed with ice-cold RIPA lysis buffer and protease inhibitor cocktail (Sigma), and then homogenized by beads (bullet-blender-storm-24, Next Advance). In total, 40 μg of protein from the samples were mixed and resolved in 4× Laemmli Sample Buffer (BioRad) and incubated in RT for 1 h before loading on polyacrylamide gels (BioRad). The electrophoresed proteins were transferred to PVDF membranes via Trans-Blot Turbo Transfer System (BioRad). The membranes were incubated for 60 min with 3.5% dry milk (Santa Cruz) in Tris-buffered saline (TBST) to block nonspecific binding sites. The membranes were incubated with the primary antibodies at 4 °C. The primary antibodies used included: anti-Cx40 and anti-Cx43 (both at 1:1000 in 5% BSA in TBST, AB-1726 and SC-9059, respectively, Santa Cruz), and anti-β actin (1:1000 in 5% BSA in TBST, ab8224, Santa Cruz). This was followed by incubation with HRP-coupled secondary antibodies for 2 h at room temperature (BioRad, 1:5000 in 3.5% milk in TBST). Proteins were revealed with ECL Prime (BioRad), and images were acquired using a LAS4000 Camera (GE Healthcare). For quantifications, band intensities were measured, and densitometry was determined using imageJ. Unprocessed scans of the resulting blots are available in the Source Data file.

**Laser-confocal optical action potentials recordings**. EHTs were loaded with the voltage-sensitive dye, FluoVolt (Invitrogen) and imaged using the line-scan mode of Zeiss LSM-710 laser-scanning confocal microscope. Tissues were excited using a 488-nm solid-state laser, and emission was split by a variable secondary dichroic beam splitter set at 493 nm collecting the high wavelengths part of the spectrum ($\lambda > 493$ nm) into the photomultiplier detector. The tissues were kept in a heated organ bath in Tyrode's solution, and field stimulated (Warner-Instruments stimulator). The acquired optical action potentials depict changes in fluorescence that is emitted by the cells, which correlates with changes in membrane potential. Such recordings are noisier than the direct electrical intracellular patch–clamp recordings, and lack absolute values. A custom-written Matlab software was used for signal analysis of the optical signals and measurement of APD at 30, 50, and 90% of repolarization ($APD_{30}$, $APD_{50}$, and $APD_{90}$).

**Optical mapping**. To study the tissues' conduction properties, EHTs were incubated with 25 μM of the voltage-sensitive dye Di-4-ANBDQBS (purchased from Prof. Leslie M. Loew, University of Connecticut Health Campus) for 10 min at room temperature. Imaging was performed using a fast electron-multiplying charge coupled device (EM-CCD) camera (Evolve® 512 Delta, Photometrics, 512 × 512 pixel) mounted on a fluorescent macroscope (Olympus MVX10). EHTs were maintained in a heated organ bath in Tyrode's solution and electrically point-stimulated using a custom-built platinum/iridium bipolar electrode. The tissues were excited using LEDs (X-Cite® TURBO, Excelitas-Technologies) with peak wavelengths at 630 nm, and emission was passed through 660 -nm long-pass dichroic mirror and 665 -nm long-pass filter (Chroma). A custom-designed software (kindly provided by Prof. Bum-Rak Choi, Brown University) was used for analysis, generation of activation maps, and conduction velocity measurements[50]. All optical experiments were performed after initial incubation with Blebbistatin for 30 min (5 μmol/L, Sigma-Aldrich) to prevent motion artifacts. Optical mapping studies were performed during spontaneous rhythm, during arrhythmias, and during pacing (point stimulation using a custom-mode electrode). For quantifying AP (APD) and conduction (CV) properties EHTs were mapped during point stimulation at a fixed rate (1 Hz). To derive restitution APD and CV plots, EHTs were paced at escalating frequencies (several steps from a pacing CL of 1000 ms to 200 ms).

**Effective refractory period measurements**. Programmed electrical stimulation was used to measure the effective refractory periods (RPs) of the atrial and ventricular EHTs. These experiments involved gradual shortening of the coupling interval (S1–S2) between the last stimulus of a train of pacing stimuli given at a fixed rate (S1, pacing CL of 500 ms) and a premature extra-stimulus (S2) until loss of capture of the premature stimuli.

**Pharmacological studies**. Vernakalant (30 μM, Cardiome), carbomylcholin (2, 5, and 10 μM), lidocaine (100 μM), flecainide (10 μM), and isoproterenol (10 μM) were dissolved in $H_2O$, and nifedipine (0.1 μM) was dissolved in DMSO (all from Sigma-Aldrich). Optical AP recordings and force measurements were performed 10 min after addition of the tested drugs.

**Force measurements**. EHTs were placed on stainless steel hooks attached to a force-transducer and a length controller (Aurora-Scientific, model 403 A/322 C), perfused with Tyrode's solution, and electrically field stimulated. Following initial length standardization, EHTs were stretched at sequential steps of 0.1 mm up to 124% of the initial length. Both the passive and active forces generated by the tissues were continuously measured. Forces were also measured at different extracellular $Ca^{2+}$ levels (0.2–3.2 mM, 2 Hz pacing), following application of iso-proterenol (10 μM) and nifedipine (0.1 μM), and at various pacing frequencies (1–2.5 Hz). A perfusion system (Warner-Instruments) was used to maintain temperature at 31 ºC and infuse drugs. Force and length signals were digitally recorded and analyzed using custom-written Matlab software. At the end of the experiments, the EHTs cross-sectional area (CSA) was measured for normalization.

**Statistical analysis**. GraphPad Prism 6 was used for statistical analysis. Continuous variables are presented as mean ± SEM. Categorical variables are expressed as frequencies. Normality tests (d'agostino & pearson or shapiro wilk) were performed for the continuous variables. For single-cell (patch–clamp) data and for the confocal optical imaging studies of the EHTs, samples size ($n$) represents the number of cells analyzed from ≥ three independent experiments or EHTs.

Repeated-measurements two-way ANOVA followed by Sidak post hoc analysis was carried out for comparison of serial measurements in multiple groups such as between the atrial and ventricular EHTs in the: (1) dose–response pharmacological studies; (2) for the restitution plot studies evaluating changes in CV and $APD_{90}$ at different pacing rates; and (3) for evaluating changes in the active forces measured at different EHT lengths. Unpaired Student's $t$ test was carried out for comparisons between two groups of continuous variables. For cases where the normal distribution criterion was not met, the Mann–Whitney test was used. For categorical variables, the Fisher test was performed.

We considered $p < 0.05$ to be statistically significant.

**Reporting summary**. Further information on research design is available in the Nature Research Reporting Summary linked to this article.

## Data availability
The data that support the findings of this study are available within the article and its supplementary information files or from the corresponding author upon reasonable request. The source data underlying most figures (Figs. 1–4, 6; Supplementary Figs. 2, 3) are provided as a Source Data file.

## Code availability
The custom-written Matlab software used in this study is available at: https://zenodo.org/badge/latestdoi/224485637.

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

## Acknowledgements

This study was funded by the European Research Council (ERC-2017-COG-773181-iPS-ChOp-AF) and by the Technion-UHN research collaboration fund.

## Author contributions

I.G. and L.G. designed the experiments; S.P., Y.N. and G.K established the hPSC atrial and ventricular differentiation system and provided the hPSC chamber-specific cardiomyocytes for the study. I.G. derived the chamber-specific EHTs, performed the detailed characterization studies (immunostaining, gene expression, and electrophysiological studies) and also analyzed the results. A.S., A.G., N. Shaeen established the methods and performed some of the optical electrophysiological studies. N. Setter performed the western blot experiments. I.G. and L.G. wrote the paper. All authors read and approved the paper, and L.G. supervised this research project.

## Competing interests

G.K. is a scientific cofounder and advisor of BlueRock Therapeutics. S.P. has an active consulting agreement with BlueRock Therapeutics. The remaining authors declare no competing interests.
