## [Peer Review File · Nature Communications]

Reviewers' Comments:

Reviewer #1:

Remarks to the Author:

This manuscript presents data characterizing two types of engineered heart tissues (EHT) built from cells, which were differentiated from the human embryonic stem cells into two different phenotypes. The authors conclude that these two EHT phenotypes correspond to mature human atrial and ventricular phenotypes, and thus can serve as models of human heart diseases. The basis for this conclusion includes the following: the two different types of cells spontaneously beat at different frequencies; they have different electrophysiological and mechanical properties, and they have different levels of expression of several molecular markers. The authors are correct that numerous previous reports failed to provide compelling evidence that human embryonic or induced pluripotent stem cells can be transdifferentiated into an adult human myocyte phenotype of atrial or ventricular type. Unfortunately, this failed to provide a sufficient evidence of successful differentiation into adult human atrial or ventricular myocytes as well:

1. The major problem with the results is that the real human atrial and ventricular myocytes do not spontaneously beat as all, they maintain steady resting potential due to high level of expression of IK1. Only immature cells and cells of SA or AV nodes are capable to spontaneously beat, or myocytes from other compartments which were de-differentiated into pacemaker cells by pathological activation of various developmental transcriptional programs (i.e. Notch1, Tbx5/Ptx2, Tbx18, etc.). Moreover, lack of stable resting potentials in these cells (Figure 1d clearly shows diastolic depolarization in both "atrial" and "ventricular" action potentials, indicates their immaturity and low level of expression of IK1. Data has to be provided comparing IK1 protein and mRNA expression in real human atrial and ventricular cells/tissue and in engineered tissues. Functionally, the authors should provide values of resting potential in these cells to prove that they are mature cells.

2. Several cardiac markers were used in this study to identify and differentiate atrial versus ventricular cells: NKX2-5, CTnT, MLC2V, with the latter being the main segregation factor for ventricular versus atrial cells. It is not convincing, considering that mature atrial myocytes have numerous factors differentiating them from ventricular myocytes and vice versa. For example, atrial myocytes nearly exclusively express Connexin 40, IKCh, IKur, etc. Please present mRNA data showing the entire myocyte expression profile to demonstrate that these are indeed atrial versus ventricular mature myocytes.

3. Measurements of action potential duration are meaningless without rate at which they were recorded, because APD is strongly rate depended (Figure 1d-e). What was the stimulation rate at which action potentials were recorded? One needs to measure restitution properties of APD, which is dependence of APD from pacing cycle length, in order to demonstrate their differences. Subsequent figure 3 shows measurements at 1,5Hz, but this is not sufficient to demonstrate restitution properties. Please show the entire restitution curves APD vs PCL (pacing cycle length).

4. Figure 1e shows very slow conduction velocity that corresponds to human sinus node conduction velocity (see Fedorov et al, Optical mapping of the isolated coronary-perfused human sinus node, JACC 2010) and does not correspond to much faster atrial or ventricular conduction velocity. This again indicates that these cells are as immature as previously reported cell lines and engineered tissues derived from either pluripotent or embryonic stem cells.

5. Figure 2: It is helpful to see differences in expression of ANF and KCNJ3, however, expression of KCNJ3 in the "ventricular" cells is rather surprising. What about Cx40 and IKur? Please present bioinformatics analysis and expression maps differentiating the two EHT, not just few selected genes.

6. Figure 3 shows functional differences indicating cholinergic response in "atrial", but not

"ventricular" EHT. Unfortunately, only one concentration was selected to probe this difference, which is not sufficient. Please show expression of mRNA and protein for I_KCh.

7. Figure 4 shows optical mapping in EHTs using voltage sensitive dye. It indicates that tissues had discontinuous conduction and lines of block. Histological evaluation of the tissues will be helpful to assess presence or absence of other cell types, which could have been present in these tissues: fibroblasts, macrophages, etc. It is also important to measure cellular morphology, size, and presence of T-tubules, which are a clear indication of mature ventricular myocytes, and in some cases atrial myocytes.

8. Figure 4 shows frequency dependence of conduction velocity, which means that electrophysiology was measured at different pacing rate. Why did not you show it in previous figures for APD? Please show this data in a more generally accepted format: CV versus pacing cycle length, rather than CV vs. Frequency. What was the refractory period of these tissues?

9. Use of T-test is not appropriate. Please use one of the repeated measures approach. For example see Sullivan LM, Repeated Measures. *Circulation*. 2008; 117: 1238-1243.

Thus, the general conclusion that the authors were able to create mature atrial and ventricular engineered tissues is not supported by the data presented in the manuscript. Additional data is required to demonstrate: the stable resting potential; normal conduction velocity and action potential duration restitution properties; myocyte size and morphology including presence of T-tubules; the atrial and ventricular gene expression profiles; histological evaluation of tissue constructs.

Reviewer #2:

Remarks to the Author:

Drs. Gepstein and Keller group, which is one of the most successful team in this field, presented a manuscript exhibiting atrial or ventricular specific engineered heart tissues (EHT). They showed electrophysiological and contractile properties of these EHTs as well as protein and gene expression. The data look attractive, particularly in electrophysiological studies. However, there are several important limitations listed below.

1. Most importantly, the reviewer concerns the novelty of this study. The authors have already reported atrial specific EHT (cell sheet) in which they tested several antiarrhythmic drugs to treat atrial fibrillation model (ref. 23). The differences of the studies seem to be rather minor.
2. The ventricular specific induction protocol utilized here seems to be usual cardiac differentiation protocol. In fact, approximate 30 % of NKX+/cTnT+ cardiomyocytes were MLC2v- (Fig. 1b). Which type of cardiomyocytes were these MLC2v- cells? I would like to know the fraction of subtypes of cardiomyocytes in each differentiation protocol.
3. I would also like to know the maturity of cardiomyocytes in each protocol. Most of the atrial cells were MLC2v-, but even ventricular-like cells could be MLC2v- if they are immature. Is it possible that there is significant difference of maturation between the two types of cardiomyocytes induced by the two protocol?
4. In our hand, there are substantial fraction of nodal like cells by the usual cardiac differentiation protocol. Indeed, the authors implied the existence of this pacemaker cells (P.11, last paragraph).
5. One of the most reasonable explanation of the difference in conduction velocities is the difference of expression pattern of gap junction proteins. Please show the expression patterns of these proteins in each EHT.
6. Action potential pattern of atrial EHT looks different to that of atrial-like cells (Fig. 1d vs. Fig. 3a). Please explain this.
7. The figure legend 3c, d does not correspond to the actual figures. Please correct this.

Reviewer #3:

Remarks to the Author:

Goldfracht et al. have established a method of generating atrial and ventricular engineered heart tissues (EHTs) from human pluripotent stem cells (hPSCs). By modulating Wnt, activin-nodal, BMP, and RA signaling pathways, the authors generate distinct high purity atrial and ventricular cardiomyocyte populations. These cells are encapsulated in a collagen gel to form ring-shaped, chamber specific EHTs. The authors characterize the chamber-specific EHTs using immunostaining, RT-qPCR, tissue mechanics, and EP measurements. Optical mapping revealed that a majority of atrial EHTs exhibit re-entrant arrhythmias, while the ventricular EHTs had normal AP propagation. The authors demonstrate transient response to pharmacologic and electrical cardioversion of the atrial EHTs. Based on this, they suggest that atrial EHTs may be used to further study atrial arrhythmias and identify novel pharmacologic agents. Overall, the characterization of the EHTs is rigorous and the ability to generate chamber specific human EHTs to model arrhythmias is impactful. Prior to publication, the authors should address the following minor concerns.

1. The authors should further characterize whether the return to arrhythmia 15 minutes after treatment in atrial EHTs follows the same re-entrant circuit as the original arrhythmia, or if new patterns appear following treatment.
2. For conduction velocities (Fig 5). The authors should list and reference conduction velocities of adult atrial and ventricular myocardium in the manuscript text (and/or directly make and include these measurements).
3. For the analysis of active and passive tissue mechanics, a direct comparison of healthy ventricular and atrial tissues should be performed alongside the EHTs.
4. Minor typos and omissions: What is the size of the scale bars on images in Figure 2? Also, there is a typo at the end of the in "optical mapping" methods section (artifacts is cut-off).

Response to the Reviewers' Comments

We are grateful to the reviewers for their constructive criticism and for the opportunity to revise our manuscript. To address all of the reviewers' concerns, we conducted new experiments, re-analysed some of the original data, and modified several parts of the original manuscript. As mentioned in our detailed responses to the Reviewers' comments, we significantly amended the main and online supplementary figures and carefully revised the text of the manuscript. For easy tracking, all changes made in the text of the revised manuscript are highlighted in red.

Reviewer #1:

1. Comment: "The major problem with the results is that the real human atrial and ventricular myocytes do not spontaneously beat as all, they maintain steady resting potential due to high level of expression of IK1. Only immature cells and cells of SA or AV nodes are capable to spontaneously beat, or myocytes from other compartments which were de-differentiated into pacemaker cells by pathological activation of various developmental transcriptional programs (i.e. Notch1, Tbx5/Ptx2, Tbx18, etc.). Moreover, lack of stable resting potentials in these cells (Figure 1d clearly shows diastolic depolarization in both "atrial" and "ventricular" action potentials, indicates their immaturity and low level of expression of IK1. Data has to be provided comparing IK1 protein and mRNA expression in real human atrial and ventricular cells/tissue and in engineered tissues. Functionally, the authors should provide values of resting potential in these cells to prove that they are mature cells."

Response: We thank the reviewer for this important comment. As suggested by the reviewer we compared the mRNA expression levels of the *KCNJ2* gene (responsible for the Kir2.1 encoded inward-rectifying potassium current, I_{K1} , in the human heart) in the chamber-specific EHTs and in control human heart tissue samples. As can be seen in Figure 2b (top panel) and also discussed in the Results (page 7, second paragraph) and Discussion (page 27, second paragraph) section of the revised manuscript, both the atrial and ventricular EHTs expressed similar mRNA levels of the *KCNJ2* gene, with the expression levels in both engineered tissue types being significantly lower than in the adult human atrial and ventricular tissue samples. In addition, based on the reviewer's suggestion we also performed studies to measure the resting membrane potential (RMP) in isolated hPSC-derived ventricular and atrial cells used for the creation of the chamber-specific EHTs. The average RMP prior to creating the EHTs was -60 ± 2 mV for the ventricular cells and -63 ± 2 mV for the atrial cells. This information is now provided in page 6, first paragraph of the revised manuscript

Regarding the spontaneous beating of the atrial and ventricular EHTs, it does not necessary stem from the presence of spontaneous automaticity originating from the ventricular and atrial cells within these tissues. It should be mentioned that in both the ventricular and atrial differentiation protocols, a small percentage of the differentiating myocytes (~1% and ~5% in the examples provided in Figures 1b-c for the ventricular and atrial differentiation protocols) actually represent SA node like pacemaker cells; identified in the FACS analysis as $TnT^{(+)}/Nkx2.5^{(-)}$ cells. These TnT positive/Nkx2.5 negative cells (representing $5.9 \pm 1.0\%$ ($n=24$) and $9.1 \pm 1.0\%$ ($n=35$) of the differentiating cells in the ventricular and atrial differentiation protocols respectively in our new analysis) were previously shown to possess molecular and functional properties of SAN like pacemaker cells (Protze et al., Nat Biotechnol 2018, reference 21). Such cells were shown to be able to pace the heart following

transplantation. The spontaneous beating of the chamber-specific EHTs (albeit at a relatively slow rates), could therefore stem from the continuous automaticity of these SAN-like pacemaker cells, since only a small number of pacemaker cells is required to electrically activate the tissues. These results and the potential implication for the presence of such SAN like cells for driving the automaticity of the EHTs are now described in the revised manuscript (page 5, second paragraph, Figures 1b-c and corresponding legends).

Finally, it should be mentioned that the aim of our study was not to achieve maturation of the hPSC-derived cardiomyocytes but rather to derive chamber-specific (atrial or ventricular) tissues in order to advance the field forward. The ability to derive tissue-engineered cardiac muscle from hPSC has been highlighted in numerous publications. The generated engineered tissues were demonstrated to be extremely useful, in these studies, for several types of pathophysiological studies, disease modelling, and drug testing applications despite the relatively early-stage phenotype of these tissues. Importantly, a major limitation of the generated tissues in these numerous studies has been the use of a mixed population of cardiomyocytes. The current study represents a significant step in moving the field forward by demonstrating the ability to generate either ventricular or atrial specific engineered tissues with unique molecular, mechanical, and electrophysiological properties. Some examples for the importance of these models are highlighted in our study including the ability to establish a unique model of atrial fibrillation (AF) and the ability to test for atrial (or ventricular) selectivity in the effects of various drugs. The latter is of great importance for developing new drugs for AF that will affect the electrophysiological properties of the atrial tissue but will have minimal toxicity (in terms of pro-arrhythmia) in the ventricular tissue. Finally, we hope in the future studies to translate some of the significant efforts that are currently made in the field to enhance the maturation of hPSC-derived cardiomyocytes also to promote the maturation of our specific atrial and ventricular tissue models. The issue of tissue maturity is now presented and discussed to a greater detail in the revised manuscript (page 27, second paragraph).

2. Comment: "Several cardiac markers were used in this study to identify and differentiate atrial versus ventricular cells: NKX2-5, CTnT, MLC2V, with the latter being the main segregation factor for ventricular versus atrial cells. It is not convincing, considering that mature atrial myocytes have numerous factors differentiating them from ventricular myocytes and vice versa. For example, atrial myocytes nearly exclusively express Connexin 40, IKCh, IKur, etc. Please present mRNA data showing the entire myocyte expression profile to demonstrate that these are indeed atrial versus ventricular mature myocytes."

Response: Based on the reviewer's suggestion, we significantly extended the molecular comparison between the atrial and ventricular EHTs. To this end, we first performed a detailed literature search and identified molecular markers that are known to be differentially expressed between atrial and ventricular myocytes (new references: 1, 8, 40-43). We only chose markers that are known from the literature to display markedly different expression levels between the two tissue types. We then performed real-time qPCR experiments to compare the expression levels of the selected markers between the ventricular and atrial hPSC-derived EHTs. To further verify that these markers are indeed expressed differently between human atrial and ventricular tissues, we also evaluated their expression levels in RNA samples derived from control human adult atrial and ventricular tissues.

The results of these studies are presented in Figure 2b of the revised manuscript and show significant differences in the expression levels of the chamber-specific genes between the atrial and ventricular EHTs. Thus, the atrial-specific genes *GJA5* (encoding for the gap junction protein connexin 40), *KCNA5* (responsible for the expression of the ultra-rapid potassium current (I_{Kur}) in atrial cells), *KCNJ3* (responsible for the expression of the I_{KACH} potassium current in atrial cells), *NNPA* (encoding for atrial natriuretic factor), *MYL7* (encoding for the myosin regulatory light chain 2, atrial isoform) and *NR2F2* (encoding for the COUP transcription factor 2 known to play an important role in determining atrial identity) were all expressed significantly higher in the **atrial EHTs** as compared to the ventricular EHTs. These genes were also expressed significantly higher in the control human atrial tissue as compared to the control human ventricular tissue. In contrast to the atrial-specific gene expression, the expression levels of the ventricular specific markers *MYL2* (encoding for the Myosin regulatory light chain 2, ventricular isoform), *MYH7* (encoding for the beta-myosin heavy chain) and *HEY2* (a cardiac-specific transcription factor) were significantly higher in the **ventricular EHTs** when compared to the atrial EHTs; in correlation with their different levels of expression in the control adult human heart derived atrial and ventricular samples.

Importantly, in addition to differences observed at the mRNA levels we also demonstrated a significant difference in the expression of chamber-specific markers at protein level. These studies, performed either by co-immunostaining experiments or by western blot analysis, confirmed the expression of the atrial-specific proteins sarcolipin (Figure 2d) and connexin 40 (Figure 2e) primarily by the atrial-EHTs and the ventricular-specific proteins such as MLC-2v (Figure 2c) and to a certain extent (see detailed discussion below) connexin 43 (Figure 2e) primarily by the ventricular EHTs.

The information regarding these new gene expression and protein studies are presented in the Methods (page 30, first paragraph and page 31, last paragraph), Results (page 7, last two paragraphs to page 9, last paragraph and Figures 2b-e) and Discussion (page 22, last paragraph to page 23, first paragraph) sections of the revised manuscript.

3. Comment: "Measurements of action potential duration are meaningless without rate at which they were recorded, because APD is strongly rate depended (Figure 1d-e). What was the stimulation rate at which action potentials were recorded? One needs to measure restitution properties of APD, which is dependence of APD from pacing cycle length, in order to demonstrate their differences. Subsequent figure 3 shows measurements at 1,5Hz, but this is not sufficient to demonstrate restitution properties. Please show the entire restitution curves APD vs PCL (pacing cycle length)."

Response: We completely agree with the reviewer's comment that APD values are highly dependent on the beating rate. Importantly, AP recordings (and consequentially APD measurements) were performed in all studies during a fixed pacing stimulation rate. The pacing frequency in most studies were either 1Hz or 1.5Hz (mostly for the carbamylcholine and verapamil studies) and are now detailed throughout the revised paper (page 5, last paragraph and legend to Figure 1; page 11 last paragraph, page 12 second paragraph and the legend for Figure 3; page 16, first paragraph and the legend for Figure 4).

Based on the reviewer suggestion, we performed detailed pacing studies in both the ventricular and atrial EHTs, in which the tissues were paced at escalating rates (using multiple pacing CLs from a CL of 1000 to 200 ms). This allowed the generation of detailed

restitution curves depicting the changes in APD values as function of the pacing cycle-length in both EHT types. Note in the resulting APD restitution curve (Figure 4f, right panel and page 16, first paragraph), the typical APD shortening with higher pacing frequencies in both the ventricular and atrial EHTs and the significantly shorter APD values in the atrial, as compared to the ventricular, EHTs at all pacing frequencies. In a similar manner, restitution curves were also generated to evaluate for changes in conduction velocity (CV) values as function of the pacing cycle-lengths. These studies revealed the expected slowing of conduction at fast rates in both ventricular and atrial EHTs and the faster CV values in the ventricular EHTs (Figure 4f, left-panel and page 16, first paragraph).

4. Comment: "Figure 1e shows very slow conduction velocity that corresponds to human sinus node conduction velocity (see Fedorov et al, Optical mapping of the isolated coronary-perfused human sinus node, JACC 2010) and does not correspond to much faster atrial or ventricular conduction velocity. This again indicates that these cells are as immature as previously reported cell lines and engineered tissues derived from either pluripotent or embryonic stem cells."

Response: We agree with the reviewer that the conduction velocity (CV) measured in our engineered tissues are somewhat slower than adult ventricular and atrial tissues and may be closer to sinus node tissue conduction (especially for the atrial EHTs). Nevertheless, please note that the CV values that were measured (especially for the ventricular EHTs) are much faster than measurements made in most previously-described hPSC-derived cardiac tissue models (2D monolayers and 3D embryoid bodies; see new references 48,50-51) and were similar to most other 3D engineered heart tissues models studied to date (see references 27-28).

Moreover, the CV values measured in our ventricular EHT model, which can reach values that are greater than 20 cm/sec at physiological rates (Figure 4f in the revised manuscript), are in the same magnitude and not that far from those reported in the *in vivo* human heart. The CV measured in the *in vivo* in the human ventricle depends on the directionality (longitudinal, transvers, or transmural conduction), with the longitudinal CV ranging from 50-100 cm/sec in different studies. For example, Aras et al. (Circ Arrhyth Electrophysiol, 2018, reference 54) measured, using panoramic optical mapping of coronary-perfused human left ventricular wedge preparation, an average longitudinal CV of 78 cm/s, and average transverse and transmural conduction velocities of 28 cm/s and 34 cm/s respectively. The CV values in the atria are also dependent on the specific atrial structure (for example conduction along the cristae terminalis is significantly faster than other parts of the atria) and can reach a value of up to ~50 cm/sec [Hansson A, et al., Eur Heart J, 1998 (reference 52) and Li et al. Plos One 2014 (reference 53)].

In the revised manuscript we further discuss the CV measured in our models in reference to previously described hPSC-derived cardiac tissue models and in comparison to the *in vivo* human atria and ventricles (page 24, last two paragraphs). We also discuss future directions and approaches that can be used to even further improve conduction in our models (page 27, second paragraph and references 66-69).

5. Comment: "Figure 2: It is helpful to see differences in expression of ANF and KCNJ3, however, expression of KCNJ3 in the “ventricular” cells is rather surprising. What about Cx40 and IKur? Please present bioinformatics analysis and expression maps differentiating the two EHT, not just few selected genes."

Response: As discussed in the detailed response to Comment 2, we have significantly expanded the gene expression analysis and noted significant differences in the expression of different chamber-specific genes between the atrial and ventricular EHTs models. As discussed above, these gene-expression differences correlated also with the differential expression of the same genes between control adult human atrial and ventricular tissue specimens. With regards to the **IK_{ur} current**, in a similar manner to the other atrial-specific genes, we observed in our new qPCR studies a significantly higher levels of expression of the relevant gene (*KCNA5*) in the atrial EHTs as compared to the ventricular EHTs (Figure 2b, middle panel). A similar difference in gene expression was also noted between the control adult heart derived atrial and ventricular specimens (Figure 2b). With regards to the *KCNJ3* gene (responsible for the **IK_{ACh} current**), its expression level in the ventricular EHTs was minimal and markedly lower than in the atrial EHTs in a similar manner to its minimal (but existing) expression levels in the adult heart derived control ventricular tissue (Figure 2b, middle panel).

The reviewer also raises an important point regarding the types of gap junction proteins expressed by the EHTs, since they can also represent excellent chamber-specific differentiating markers. We therefore evaluated in the atrial and ventricular EHTs the expression levels of connexin 43 (Cx43, known to be expressed *in vivo* in both ventricular and atrial tissues) and connexin 40 (Cx40, known to be expressed in atrial, but not in ventricular, cells). As noted in the relevant qPCR studies (Figure 2b of the revised manuscript), the expression of the *GJA5* gene (encoding for Cx40) was significantly higher in the atrial EHTs in comparison to its expression levels in the ventricular EHTs; in a similar manner to its differential expression in the control atrial and ventricular adult cardiac tissues (Figure 2b, middle panel). In contrast, the *GJA1* gene (encoding for Cx43) was expressed at the mRNA levels in both ventricular and atrial EHTs (Figure 2b, top panel). Interestingly, its expression levels were somewhat higher in the ventricular EHTs (albeit not reaching statistical significance). A similar *GJA1* gene expression pattern was also noted in the control mRNA derived from the adult human atrial and ventricular heart tissues (Figure 2b).

To further evaluate gap junction protein (connexins) expression patterns also at the protein level we performed western blot analysis of the atrial and ventricular EHTs using antibodies targeting either Cx40 or Cx43. The results of these experiments (depicted in Figure 2e of the revised manuscript) demonstrated marked differences in the Cx40 protein levels between the atrial and ventricular EHTs. Thus, while the atrial EHTs expressed high protein levels of Cx40, the expression level of this protein in the ventricular EHTs was miniscule ($p < 0.001$). In contrast to the Cx40 expression, we found that Cx43 was expressed at the protein levels in both the ventricular and atrial EHTs (Figure 2e). Interestingly, similar to the mRNA expression pattern and to the known expression levels in the *in vivo* adult heart, the level of Cx43 protein expression was higher ($p < 0.01$) in the ventricular EHTs when compared to the atrial specimens Figure 2e. Among other reasons, these findings may contribute to the faster CV observed in the ventricular EHTs

The information regarding the aforementioned new gene expression and protein studies are presented in the Methods (page 30, first paragraph and page 31, last paragraph), Results

(page 7, last two paragraph to page 8, first paragraph; page 9, second paragraph; and Figures 2b,e), and Discussion (page 23, first paragraph and page 24, first paragraph) sections of the revised manuscript.

6. Comment: "Figure 3 shows functional differences indicating cholinergic response in “atrial”, but not “ventricular” EHT. Unfortunately, only one concentration was selected to probe this difference, which is not sufficient. Please show expression of mRNA and protein for IKACH."

Response: We thank the reviewer for his comments and suggestions. As discussed in the response to Comment 2, among the different atrial-specific genes we also evaluated the expression levels of the *KCNJ3* gene, responsible for the I_{KACH} current, in both the atrial and ventricular EHTs. As can be seen in the Figure 2b (middle panel) of the revised manuscript, the expression levels of *KCNJ3* were markedly higher in the atrial EHTs as compared to the ventricular EHTs. A similar chamber-specific differential expression pattern was also noted in the control (adult heart derived) atrial and ventricular tissues.

To address the reviewer suggestion, we also performed new experiments to assess the effects of escalating concentrations of carbonylcholine on the action-potential properties in the atrial and ventricular EHTs. The resulting dose-response curves for both EHT types are presented in Figure 3c (ii) of the revised manuscript. These studies revealed that carbonylcholine application resulted in a **dose-related** APD shortening in the atrial EHTs. In contrast application of the drug at different concentrations did not alter APD in the ventricular EHTs. This information is now provided in the Methods (page 33, third paragraph), Results [page 12, second paragraph and Figure 3c (ii)], and Discussion (page 23, second paragraph) sections of the revised manuscript.

7. Comment: Figure 4 shows optical mapping in EHTs using voltage sensitive dye. It indicates that tissues had discontinuous conduction and lines of block. Histological evaluation of the tissues will be helpful to assess presence of absence of other cell types, which could have been present in these tissues: fibroblasts, macrophages, etc. It is also important to measure cellular morphology, size, and presence of T-tubules, which are a clear indication of mature ventricular myocytes, and in some cases atrial myocytes.

Response: Most of the maps in Figures 4a-d represents various arrhythmias with multiple reentry circuits that are characterized by lines of block and other conduction abnormalities. Conduction during point stimulation (such as depicted in Figures 4e and 4g) was relatively homogeneous.

As suggested by the reviewer, we examined the presence of different cell types in the engineered heart tissues. Co-immunostainings were performed for cardiac troponin-I (cTnI) and for vimentin to identify cardiomyocytes and fibroblasts respectively within both the atrial and ventricular EHTs. As can be seen in Supplementary Figures 1a-b of the revised manuscript, these immunostainings revealed the presence of only small number of fibroblasts within both EHT types. This finding is consistent with the high percentage of cardiomyocytes in the initial cell-population that was used to create the EHTs. In a similar manner immunostaining for CD31 (a surface marker for endothelial cells) and CD163 (a marker for macrophages) revealed almost no expression of these cells in the atrial and ventricular EHTs. This information is now described in the Methods (page 30, second paragraph) and Results (page 9, first paragraph and Supplementary Figures 1a-b) sections of the revised manuscript.

We also thank the reviewer for his suggestion to measure cellular morphology and size in the ventricular and atrial EHTs. However, despite many attempts, because of the lack of clear demarcation of the cellular borders of the cells in the slices derived from the densely-packed cells in the 3D EHT, it was impossible to quantify such parameters. Because of similar limitations, we also encountered a problem in our attempts to stain for the T-tubule markers using anti-WGA and anti-JP-2 antibodies. While the immunosignal in the ventricular EHTs seemed positive, we were not confident in its degree of organization in cells within the EHTs. Consequentially, we could not determine with certainty whether organized T-tubules developed within the EHTs and therefore did not comment on this type immunostaining in the revised manuscript. It should be mentioned that in the EM studies we could not identify the presence of T-tubules.

8. Comment: Figure 4 shows frequency dependence of conduction velocity, which means that electrophysiology was measured at different pacing rate. Why did not you show it in previous figures for APD? Please show this data in a more generally accepted format: CV versus pacing cycle length, rather than CV vs. Frequency. What was the refractory period of these tissues?

Response: We thank the reviewer for this comment. As suggested by the reviewer, we now present in the revised manuscript (Figure 4F) both the CV and APD restitution plots. Note the typical slowing of conduction and APD shortening with faster pacing rates (shorter pacing cycle-lengths) in both EHT types. The CV and APD values in these restitution plots are presented as function of the pacing CL (instead of as function of pacing frequency as was presented in the original version of the manuscript).

Based on the reviewer's valuable suggestion, we also performed new pacing experiments to measure the corresponding refractory periods (RPs) of the atrial and ventricular EHTs. These experiments involved gradual shortening of the coupling interval (S1-S2) between the last stimulus of a train of pacing stimuli given at a fixed rate (S1) and a premature extra-stimulus (S2) until loss of capture of the premature stimuli [see examples of captured vs. not-captured S2's in Supplementary Figure 2a of the revised manuscript]. The results of these studies (Supplementary Figure 2b) revealed longer effective refractory period values in the ventricular EHTs in comparison to similar measurements made in the atrial EHTs (240.0 ± 9.4 vs. 86.0 ± 2.9 ms, $n = 10$, $p < 0.0001$). The information regarding the new refractory period measurements and their implications are provided in the Methods (page 33, second paragraph and Supplementary Figure 2a), Results [page 16, last paragraph to page 17, first paragraph; and Supplementary Figures 2a-b], and Discussion (page 25, second paragraph) sections of the revised manuscript.

9. Comment: "Use of T-test is not appropriate. Please use one of the repeated measures approach. For example see Sullivan LM, Repeated Measures. *Circulation*. 2008;117:1238-43."

Response: We thank the reviewer for his comment. We therefore repeated the statistical analysis using repeated measures approaches for comparisons of serial measurements made in multiple groups. These includes for example the comparisons between the atrial and ventricular EHTs in the: (1) dose-response pharmacological studies; (2) for the restitution plot studies evaluating changes in CV and APD₉₀ at different pacing rates; and (3) for evaluating changes in the active forces measured at different EHT lengths. This information is now provided in the revised statistical descriptions in the Methods section (page 34, last paragraph) and in the relevant sections of the Results and Figure Legends.

Reviewer #2

1. Comment: "Most importantly, the reviewer concerns the novelty of this study. The authors have already reported atrial specific EHT (cell sheet) in which they tested several antiarrhythmic drugs to treat atrial fibrillation model (ref. 23). The differences of the studies seem to be rather minor."

Response: We thank the reviewer for citing our previous work (Laksmann et al., reference 22). In that study, atrial cells derived from hESC were cultured as two-dimensional monolayers and their electrophysiological properties and the effects of the anti-arrhythmic drugs flecainide and dofetilide were evaluated using optical mapping and voltage-sensitive dyes. The current work differs significantly from this previous study and presents many additional novel concepts that should, therefore, significantly advance the field forward. Some of the advancements presented in the current study include:

- (1) Detailed evaluation of conduction and arrhythmogenesis (reentry circuits) in three-dimensional engineered heart tissue. The added value of studying reentry in 3D tissues beyond the previously described 2D model (as presented in the Laksmann paper) is the different biophysical conditions associated with the two conditions. For example, reentry in the 2D model (manifested as rotors) is significantly influenced by the boundary conditions, a property that is less relevant in the 3D model. Consequentially, the 3D setting may be more clinically relevant than the 2D model.
- (2) An important implication of the study is the ability to utilize the new chamber-specific EHT models to study re-entrant arrhythmias, and specifically atrial arrhythmias such as atrial fibrillation (AF) in 3D context. Interestingly, re-entrant arrhythmias developed spontaneously in most atrial-EHTs. In contrast, the ventricular EHTs seldom developed arrhythmia spontaneously or following pacing induction protocols. This difference in arrhythmias susceptibility probably stems from the marked differences observed between the two tissue types in the tissue wavelength (WL), which is a product of the tissue refractory period (or its surrogate marker APD) and CV. The atrial EHTs were characterized by a markedly shorter WL as a consequence of the slower CV and shorter refractory period and APD; a property that is associated with a greater probability for initiation and maintenance of re-entrant circuits. This finding may also explain why atrial fibrillation and other atrial arrhythmias are much more common clinically and easier to induce than ventricular arrhythmias despite the much larger ventricular tissue mass. It is interesting to note that the arrhythmias generated in the atrial EHTs were complex, often consisting of multiple simultaneous wavelets, which may resemble AF to a greater extent than the 2D model, which was characterized by a single reentrant circuit.
- (3) Detailed anti-arrhythmic drug studies demonstrating the ability to identify chamber-specific electrophysiological effects of the studied pharmacological agents. This was exemplified by the atrial-specific effects of verapamil in prolonging atrial APD and by the greater effects of Lidocaine in slowing conduction in the ventricular tissue. The ability to identify the potentially diverse and different effects of drugs on the atrial and ventricular tissues is important since a major goal in developing novel therapies for atrial fibrillation, for example, is in identifying agents that will robustly affect the atrial electrical activity but will possess minimal effects on the ventricular electrophysiological properties in order to limit potential ventricular pro-arrhythmia. In the Laksmann study pharmacological agent with potential chamber-specific actions were not tested
- (4) Demonstrating the ability of using the atrial EHT model as a novel atrial arrhythmia model for drug testing in order to screen the ability of drugs (flecainide and vernakalant)

or other interventions (electrical "shock") to terminate the reentrant activity ("cardioversion") and to prevent the development of new arrhythmias ("rhythm control" approach).

- (5) Evaluate the contractile properties of the atrial and ventricular EHTs at different loading conditions (which cannot obviously be studied in 2D tissues). These studies demonstrated different contractile properties of the atrial and ventricular EHTs, with regard to the magnitude of the forces generated and their calcium sensitivity properties. Interestingly, the chamber-specific difference recapitulated known difference in the same properties between atrial and ventricular muscle as previously described in the in vivo human heart.
- (6) Detailed analysis showing chamber-specific differences in the expression of several genes and proteins (with specific emphasis on the type of gap junction proteins expressed) between the atrial and ventricular EHTs,

2. Comment: "The ventricular specific induction protocol utilized here seems to be usual cardiac differentiation protocol. In fact, approximate 30 % of NKX⁺/cTnT⁺ cardiomyocytes were MLC2v⁻ (Fig. 1b). Which type of cardiomyocytes were these MLC2v⁻ cells? I would like to know the fraction of subtypes of cardiomyocytes in each differentiation protocol."

Response: As described in detail in the paper by Lee, Sprotze, et al. (Cell Stem Cells 2017; Reference 8) the ventricular-specific differentiation protocol, also used in the current study, differs from the regular cardiac differentiation scheme and leads to the derivation of a purified population of cells with a ventricular-specific phenotype.

Based on the Keller's group experience, the FACS analysis targeting MLC-2V tends to underestimate the number of cells that are NKX⁺/cTnT⁺/MLC-2V⁺ (thereby overestimating the percentage of NKX⁺/cTnT⁺ cardiomyocytes that are MLC-2V negative). Consequentially, to evaluate more accurately the actual incidence of the NKX⁺/cTnT⁺/MLC-2V⁺ cell population among the differentiating cells derived using the ventricular-specific differentiation protocol, we performed two types of new studies:

(1) The differentiating hPSC-derived ventricular cardiomyocytes were plated as single cells. After 7 days in culture, we performed co-immunostaining studies of the plated cells for cardiac troponin I (cTnI) and MLC-2v. The resulting images were analysed using the Imaris software. Importantly these studies revealed that 87.3±0.1% of the cTnI positive cells also expressed MLC-2V. This information is now provided in the Methods (page 30, last paragraph to page 31, first paragraph) and Results (page 5, first paragraph) sections of the revised manuscript.

(2) We also performed quantitative analysis of ventricular EHTs that were sliced and then co-immunostained with antibodies targeting cTnI and MLC-2V. Volumetric analysis of the resulting immunostainings revealed that 88.8±2.7% of the cTnI positive cellular volume was also positive for the MLC-2v immunosignal. This information is now provided in the Methods (page 30, second paragraph) and Results (page 8, third paragraph) sections of the revised manuscript.

3. Comment: "I would also like to know the maturity of cardiomyocytes in each protocol. Most of the atrial cells were MLC2v⁻, but even ventricular-like cells could be MLC2v⁻ if they are immature. Is it possible that there is significant difference of maturation between the two types of cardiomyocytes induced by the two protocol?"

Response: We thank the reviewer for his comment and question. As discussed in the response to Comment 2, almost 90% of the differentiating cells in the ventricular differentiation protocol were MLC-2V positive by eventual immunostaining. To further examine and compare the degree of maturation of the hPSC-derived atrial and ventricular cardiomyocytes within the chamber-specific EHTs, we performed qPCR analysis to quantify the expression levels of three general cardiomyocyte-specific genes (*TNNI1*, *TNNT2*, and *KCNJ2*) that change their level of expression as function of the maturation status. The results of these studies [Figure 2b (upper-panel) of the revised manuscript] revealed similar levels of expressions of these genes between the atrial and ventricular EHTs, indicating a similar degree of maturation. In addition to the gene expression studies, we also measured the resting membrane potential (RMP) of the hPSC-derived atrial and ventricular cells (prior to their use for creation of the EHTs) since this value may also correlate with the degree of cell maturation. These patch-clamp studies demonstrated similar RMP values between the atrial ($-63\pm 2\text{mV}$) and ventricular ($-60\pm 2\text{mV}$) cells. Finally, transmission electron microscopy images of the atrial and ventricular EHTs (Supplementary Figures 1c and 1d) showed a similar degree of ultrastructural maturation and sarcomeric organization. Thus, it seems that the degree of maturation does not seem to differ significantly between the hPSC-derived atrial and ventricular cells and consequentially this factor probably is not the reason for the phenotypic differences observed between the atrial and ventricular EHTs.

The issues relating to the comparison of the degree of maturation of the differentiating chamber-specific cardiomyocytes are now presented in the revised manuscript (page 6, first paragraph; page 7, second paragraph; page 11, first paragraph; Figure 2b and Supplementary Figures 1c-d).

4. Comment: "In our hand, there are substantial fraction of nodal like cells by the usual cardiac differentiation protocol. Indeed, the authors implied the existence of this pacemaker cells (P.11, last paragraph)."

Response: According to our previous study (Protze et al., Nature Biotechnology 2017; reference 22) nodal like cells can be identified during hPSC differentiation as NKX2.5 negative / troponin positive cells. Based on this classification, we re-analysed our data and quantified, using FACS analysis, the fraction of SA nodal like cells obtained in the ventricular differentiation protocol. This cell population can be readily identified in the FACS analysis as the $\text{TnT}^{(+)}/\text{Nkx2.5}^{(-)}$ cells and account, in the examples provided in Figures 1b-c, for ~1% and ~5% of the differentiating cells in the ventricular and atrial differentiation protocols respectively. Summarizing several differentiation experiments we found that the SA node like cells represent $5.9\pm 1.0\%$ ($n=24$) of the differentiating cells in the ventricular differentiation protocol and $9.1\pm 1.0\%$ ($n=35$) in the atrial protocol. Consequentially, the presence of such SAN like pacemaker cells in the population used to generate the EHTs may be responsible for the spontaneous automaticity (albeit at slow rate) observed in both types of EHTs. These results are now described in the revised manuscript (page 5, second paragraph, Figures 1b-c and corresponding legends).

5. Comment: "One of the most reasonable explanation of the difference in conduction velocities is the difference of expression pattern of gap junction proteins. Please show the expression patterns of these proteins in each EHT."

Response: The reviewer raised an important point. As suggested by the reviewer, we examined the expression patterns of connexin 43 (Cx43, known to be expressed by both

ventricular and atrial cells) and connexin 40 (Cx40, known to be expressed only by atrial cells) in the chamber-specific EHTs at both the mRNA and protein levels. As noted in the relevant qPCR studies (Figure 2b of the revised manuscript) the *GJA1* gene (encoding for Cx43) was expressed in both ventricular and atrial EHTs. Interestingly, its expression levels were somewhat higher in the ventricular EHTs (albeit not reaching statistical significance). In contrast, the expression of the *GJA5* gene (encoding for Cx40) was significantly higher in the atrial EHTs in comparison to its expression levels in the ventricular EHTs; in a similar manner to its different expression pattern in the control atrial and ventricular adult cardiac tissues (Figure 2b, middle panel).

More importantly, to study the connexins' gene expression patterns also at the protein level, we performed western blot analysis of the atrial and ventricular EHTs using antibodies targeting either Cx40 or Cx43. The results of these experiments (depicted in Figure 2e of the revised manuscript) demonstrated marked differences in the Cx40 protein levels between the atrial and ventricular EHTs. Thus, while the atrial EHTs expressed high protein levels of Cx40, the expression level of this protein in the ventricular EHTs was miniscule ($p < 0.001$). In contrast, similar to its mRNA expression pattern and to the known expression levels in the *in vivo* adult heart, Cx43 was expressed in both chamber-specific EHT but the level of Cx43 protein expression was significantly higher ($p < 0.01$) in the ventricular EHTs as compared to the atrial specimens Figure 2e.

Among other reasons (such as the steeper slope of phase 0 of the action potential in the ventricular cells), the aforementioned differences in connexin expression patterns may also contribute to the faster conduction velocity (CV) observed in the ventricular EHTs. Thus, the fact that the ventricular tissues contains primarily homotypic Cx43-Cx43 gap junctions, whereas the atrial tissues contain in addition to such gap junctions also homotypic Cx43-Cx43 gap junctions as well as homotypic and heteromeric gap junctions consisting of a mixture of Cx43 and Cx40 proteins may result in reduced CV in the latter (atrial) tissue. This was exemplified in an elegant study by the Kleber and Safitz group (Beauchamp et al., *Circ Res* 2006, reference 49) where they studied cultured atrial strands prepared from different combinations of control and homozygous or heterozygous mice with genetic deficiency of either Cx43 or Cx40. Their results suggest that the relative abundance of Cx43 and Cx40 is an important determinant of the tissue's impulse propagation, whereby dominance of Cx40 decreases and dominance of Cx43 increases electrical wave propagation velocity.

The aforementioned studies regarding the gap junction protein expression patterns and their implication for the chamber-specific EHT conduction is now presented in the Methods (page 31, last paragraph), Results (page 9, last paragraph and Figures 2b and 2e) and Discussion (page 24, first paragraph) sections of the revised manuscript

6. Comment: "Action potential pattern of atrial EHT looks different to that of atrial-like cells (Fig. 1d vs. Fig. 3a). Please explain this."

Response: The differences between the action potential recordings in the original Fig.1d and Fig.3a stems not only from differences in the specimens from which they were recorded (cells vs. tissues) but also from the method of recordings. Hence, in Fig.1d the tracings represent action potential recordings from isolated single-cells performed using intracellular patch-clamp recordings. In contrast in Fig.3a, the traces represent optical action potentials recordings made from atrial EHT loaded with fluorescent voltage-sensitive dyes and imaged using laser confocal microscopy. The typical patch-clamp intracellular recordings displays

the absolute changes in membrane potential, whereas the optical action potential depicts changes in fluorescence emitted by the cells correlating with changes in membrane potential. Such recordings are noisier than the direct electrical intracellular patch-clamp recordings and lack absolute values. This difference is now better explained in the revised manuscript (page 32, second paragraph). We also replaced some of the confocal images in Figure 3a with clearer traces with less noise.

7. Comment: "The figure legend 3c, does not correspond to the actual figures. Please correct this."

Response: We thank the reviewer for noting this mistake. We have corrected the relevant figure legend (Figure 3c legend in the revised manuscript).

Reviewer #3

1. Comment: "The authors should further characterize whether the return to arrhythmia 15 minutes after treatment in atrial EHTs follows the same re-entrant circuit as the original arrhythmia, or if new patterns appear following treatment."

Response: To address the reviewer's suggestion we re-analysed our data (the acquired optical mapping dynamic displays) to further characterize the reentrant arrhythmia patterns also for the cases of arrhythmia recurrences following treatment. Interestingly, in atrial-EHTs that were treated with the different drugs (flecainide and verapamil), the return to arrhythmia followed the same reentry circuit as the original arrhythmia. In contrast, in the EHTs that were treated only by applying field stimulation, the recurrent arrhythmia patterns (developing 15 minutes following termination of the original arrhythmias) were not necessarily of the same type as the original arrhythmias; as in 40.9% of the cases the arrhythmia pattern changed to a different reentry circuit. This new information is now provided in the revised manuscript (page 17, last paragraph to page 18, first paragraph).

2. Comment: "For conduction velocities (Fig 5). The authors should list and reference conduction velocities of adult atrial and ventricular myocardium in the manuscript text (and/or directly make and include these measurements)."

Response: We thank the reviewer for his suggestion and now list, reference, and discuss the conduction velocities (CVs) measured in the adult human atria and ventricles in comparison to the same measurements made in the chamber-specific EHTs. It should be mentioned that it is difficult to directly compare the *in vivo* CVs between the two chambers because of the different methods and models used and the fact that the chamber structure and directionality may impact CV values. The CV measured in the *in vivo* in the human ventricles depends on the directionality (longitudinal, transvers, or transmural conduction), with longitudinal CV ranging from 50-100 cm/sec in different studies. For example, Aras et al. [Circ Arrhythm Electrophysiol, 2018 (reference 54)] measured, using panoramic optical mapping of coronary perfused human left ventricular wedge preparation, an average longitudinal CV of 78 cm/s, and average transverse and transmural conduction velocities of 28 cm/s and 34 cm/s respectively. CV in the atria is also dependent on the specific atrial structure (for example conduction along the crista terminalis is significantly faster) and can reach a value of up to ~50 cm/sec [Hansson A, et al., Eur Heart J, 1998 (reference 52) and Li et al. Plos One 2014 (reference 53)].

Interestingly, CVs measured in our chamber-specific EHTs (and specifically in the ventricular EHTs) were significantly higher than most previous reports evaluating hPSC-derived 2D monolayer-based or 3D EB-based models (references 48, 50, 51) and were similar to other hPSC-based 3D engineered heart tissue models (references 27,28) indicating a relative high degree of tissue structural organization and electrophysiological maturation. Hence, the conduction velocity values measured in the ventricular EHT model can reach values that are greater than 20 cm/sec at physiological rates (Figure 4f in the revised manuscript). These values are in the same magnitude and are not that far from those reported in human atrial and ventricular tissue reported above.

In the revised manuscript we further discuss the measured conduction velocity values in our models in reference to previously described hPSC-derived cardiac tissue models and in comparison to the *in vivo* human atria and ventricles (page 24, both paragraphs). We also

discuss future directions and approaches that can be used to even further improve conduction in our models (page 27, second paragraph).

3. Comment: "For the analysis of active and passive tissue mechanics, a direct comparison of healthy ventricular and atrial tissues should be performed alongside the EHTs."

Response: We appreciate the reviewer suggestion. However, since we do not have access to viable human cardiac tissues we cannot perform such comparable measurements. A detailed literature search, however, identified some relevant studies. For example, a study by Van der Velden, J., et al. ["Isometric tension development and its calcium sensitivity in skinned myocyte-sized preparations from different regions of the human heart." *Cardiovasc Res.*, 1999 (reference 63)] found that the average isometric tension at saturating calcium concentration of ventricular human tissue was 48 ± 5 mN/mm². Similar to our study forces measured from the atrial human tissue (27 ± 4 mN/mm²) were smaller than in the ventricular tissue. Interestingly, in this study the concentration of calcium required for half-maximal activation was significantly higher in the atrial than in ventricular preparations. This observation correlates well with the finding of our study (see the difference between the atrial and ventricular force/calcium curves in Figure 6d of the revised manuscript).

In other studies [Bening et al. *BMC Cardiovasc Dis* 2014 (reference 61) and Bening et al, *BMC Cardiovasc Dis* 2016 (reference 60)] forces were measured only from atrial tissue samples, which are easier to obtain than ventricular samples during open heart surgery. Forces produced by these atrial tissues ranged between 2.0-4.0 mN. Similarly, Maier et al. (*Am J Physiol Heart Circ Physiol.* 2000; reference 62) also measured forces from intact human atrial trabecula and found that the amplitude of these forces ranged between 6 to 8 mN/mm².

In summary, the forces reported in the above-mentioned studies are generally higher than those found in our EHT models. The forces measured in our EHT model (~ 1 -2 mN/mm²), however, are in a similar range to other previously reported tissue engineered cardiac muscle models utilizing hPSC-CMs, with some studies reporting higher values (references 28, 57) and others lower or similar forces (references 58, 59). Interestingly, our findings in the chamber-specific EHT models recapitulated some of the conceptual differences in the mechanical properties of atrial and ventricular tissues identified in the above-mentioned human heart-derived tissue studies in terms of the magnitude of the measured forces (lower in the atrial tissues) and the nature of their calcium dependency.

This information is now provided in the Results (page 20, first to second paragraphs) and Discussion section (page 26, last paragraph to page 27, first paragraph; and references 57-62) of the revised manuscript.

4. Comment: "Minor typos and omissions: What is the size of the scale bars on images in Figure 2? Also, there is a typo at the end of the in "optical mapping" methods section (artifacts is cut-off)."

Response: We thank the reviewer for pointing these mistakes. Based on his comment, we corrected the typo (page 33, first paragraph) and also added the sizes of the relevant scale bars (20 μ m) in the legends Figures 2c-d.

Reviewers' Comments:

Reviewer #1:

Remarks to the Author:

Manuscript was significantly improved. The authors are commended for conducting new important experiments, addressing previous limitations. One important limitation remains: measured resting potentials were -60mV for ventricular and -63mV for atrial cells. Both these numbers are quite depolarized as compared to that recorded in mature ventricular and atrial myocytes, reported between -80mV and -90mV. This limitation should be acknowledged.

Reviewer #2:

Remarks to the Author:

Goldfracht, et al. worked hard to respond reviewers' comments raised in the initial round of review process, resulting in significant improvement of the manuscript. Most of my concerns were cleared; however, there have been two important papers (Lee, et al. Science 2019, PMID#31371612 and Zhao, et al. Cell 2019, PMID#30686581) published that are related to the current study during the long revision period. I still concern novelty of this current work.

Reviewer #3:

Remarks to the Author:

Thank you for addressing our concerns. No further comments. Recommend accept.

And a friendly reminder to the authors that not all reviewers are "he's"!

Response to the Reviewers' Comments

Reviewer #1

Comment: "Manuscript was significantly improved. The authors are commended for conducting new important experiments, addressing previous limitations. One important limitation remains: measured resting potentials were -60mV for ventricular and -63mV for atrial cells. Both these numbers are quite depolarized as compared to that recorded in mature ventricular and atrial myocytes, reported between -80mV and -90mV. This limitation should be acknowledged"

Response: We thank the reviewer for his suggestion and added in the discussion section (page 20, second paragraph) a comment regarding the relative depolarized resting membrane potential of the differentiated hPSC-derive atrial and ventricular cells as a marker of relative immaturity of the cells used to create the EHTs.

--

Reviewer #2

Comment: "Goldfracht, et al. worked hard to respond reviewers' comments raised in the initial round of review process, resulting in significant improvement of the manuscript. Most of my concerns were cleared; however, there have been two important papers (Lee, et al. Science 2019, PMID#31371612 and Zhao, et al. Cell 2019, PMID#30686581) published that are related to the current study during the long revision period. I still concern novelty of this current work."

Response: We thank the reviewer for this encouraging remark and also for suggesting the new references. These new reference, are now cited (references 48-49) and discussed (page 15, last paragraph and page 20 last paragraph) in the revised version

--

Reviewer #3

Thank you for addressing our concerns. No further comments. Recommend accept.

And a friendly reminder to the authors that not all reviewers are "he's"!

Response: We thank the reviewer for her/his kind words.